# Comparison of MODIS and SWAT Evapotranspiration over a Complex Terrain at Different Spatial Scales

Olanrewaju O Abiodun[1], Huade Guan[1], Vincent E.A. Post[1], Okke Batelaan[1]

[1]National Centre for Groundwater Research and Training, College of Science and Engineering, Flinders University, Australia

*Correspondence to*: Olanrewaju O Abiodun (lanre.abiodun@flinders.edu.au)

**Abstract.** In most hydrological systems, evapotranspiration (ET) and precipitation are the largest components of the water balance, which are difficult to estimate, particularly over complex terrain. In recent decades, the advent of remotely-sensed data based ET algorithms and distributed hydrological models has provided improved spatially-upscaled ET estimates. However, information on the performance of these methods at various spatial scales is limited. This study compares the ET from the MODIS remotely sensed ET dataset (MOD16) with the ET estimates from a SWAT hydrological model on graduated spatial scales for the complex terrain of the Sixth Creek Catchment of the Western Mount Lofty Ranges, South Australia. ET from both models were further compared with the coarser-resolution AWRA-L model at catchment scale. The SWAT model analyses are performed on daily timescales with a 6-year calibration period (2000-2005) and 7-year validation period (2007-2013). Differences in ET estimation between the SWAT and MOD16 methods of up to 31%, 19%,15% ,11% and 9% were observed at respectively 1 km$^2$, 4 km$^2$, 9 km$^2$, 16 km$^2$ and 25 km$^2$ spatial resolutions. Based on the results of the study, a spatial scale of confidence of 4 km$^2$ for catchment scale evapotranspiration is suggested in complex terrain. Land cover differences, HRU parameterization in AWRA-L and catchment-scale averaging of input climate data in the SWAT semi-distributed model were identified as the principal sources of weaker correlations at higher spatial resolution.

**Key words:** Evapotranspiration, MOD16, SWAT, AWRA-L, complex terrain, spatial scale

## 1  Introduction

**1  Introduction**
In most hydrological systems, evapotranspiration (ET) and precipitation are the largest components of the water
balance (Nachabe et al., 2005) and yet the most difficult to estimate particularly over complex terrain (Wilson and
Guan, 2004). In arid and semi-arid environments ET is a significant sink of groundwater with ET often exceeding
precipitation (Domingo et al., 2001;Cooper et al., 2006;Scott et al., 2008;Raz-Yaseef et al., 2012). Reliable
estimation of ET is integral to environmental sustainability, conservation, biodiversity and effective water
resource management (Cooper et al., 2006;Boé and Terray, 2008;Zhang et al., 2008a;Tabari et al., 2013).
Moreover, ET will be one of the most severely impacted hydrological components of the water cycle alongside
precipitation and runoff as a consequence of global climate change (Abtew and Melesse, 2013).
Reliable, cheap and generally accessible methods of estimating ET are essential to understand its role in catchment
processes. ET is principally measured and estimated using ground based measurement tools and/or through
various modelling techniques often involving remote sensing (Drexler et al., 2004;Tabari et al., 2013). Ground
based measurement methods such as the Bowen Ratio Energy Balance (BREB), Eddy Covariance (EC), Large
Aperture Scintillometers (LAS) and lysimeters have been regarded as the most accurate and reliable ET
determination methods (Kim et al., 2012a;Rana and Katerji, 2000;Liu et al., 2013), but they are spatially and/or
temporally limited (Wilson et al., 2001;Glenn et al., 2007). Despite the relative reliability of ground based
measurement methods, there are inherent uncertainties associated with the different methods, which affect the
accuracy of ET measurements (Baldocchi, 2003;Brotzge and Crawford, 2003;Drexler et al., 2004;Zhang et al.,
2008a). Ground based measurement methods are particularly prone to significant errors related to instrument
installation (Allen et al., 2011). Mu et al. (2011) observed that multiple EC towers on a site can have uncertainties
ranging between 10-30% and Liu et al. (2013) documented uncertainty ranges of over 27% between EC and LAS
measurements over the same site on an annual scale. EC towers have also been observed to encounter energy
balance closure challenges (Wilson et al., 2002), while other challenges of the EC method such as inaccuracies
due to complex terrains have been documented by Feigenwinter et al. (2008). Furthermore, Kalma et al. (2008),
conducted a review of 30 remote sensing ET modelling results relative to ground based measurements and
contended that the ground based measurement methods were not incontrovertibly more reliable than the remote
sensing ET modelling methods. Moreover, most of the ground based measurement methods are usually cost
intensive thereby constraining measurements over large areas and thus making spatial extrapolation difficult
(Moran and Jackson, 1991;Verstraeten et al., 2008;Melesse et al., 2009;Fernandes et al., 2012).

In more recent years, the spatial challenges associated with ET estimations are being eased by the increased availability of remotely-sensed data. The use of remotely-sensed input data in many surface energy balance algorithms and highly parameterized hydrological models have been extensively documented (Kalma et al., 2008;Hu et al., 2015;Zhang et al., 2016). The advances in remote sensing have seen these methods become prominent in water resource assessment studies (Sun et al., 2009;Vinukollu et al., 2011;Anderson et al., 2011;Long et al., 2014;Zhang et al., 2016).

Several hydrological models and remotely-sensed based surface energy balance models are currently used in ET simulations globally (Zhao et al., 2013;Chen et al., 2014;Larsen et al., 2016;López López et al., 2016;Webster et al., 2017). However, the relative accuracy of these models relative to one another should be extensively explored to improve our understanding of the ET estimation from these algorithms. Two of the more prominent ones will be comprehensively evaluated in this study at various spatial scales – The Soil and Water Assessment Tool (SWAT) (Neitsch et al., 2011) and the MODIS ET product (Mu et al., 2013) derived from remotely-sensed data from the Moderate Resolution Imaging Spectroradiometer (MODIS) instrument aboard the National Aeronautics and Space Administration (NASA) Aqua and Terra satellites. The evapotranspiration product of a third model, the Australian Water Resource Assessment model (AWRA_L) with a coarser resolution will also be evaluated at the catchment scale.

The MODIS ET (MOD16) is based on the Penman-Monteith equation, the AWRA-L uses the Penman equation, while the SWAT ET algorithm also has the Penman-Monteith equation as one of the three user-selectable methods of estimating ET. In this study, the Penman-Monteith method in SWAT is used for a direct comparison with the MOD16 and the AWRA-L.  Moreover, the Penman-Monteith equation is regarded as one of the most reliable methods for ET estimation over various climates and regions (Allen et al., 2005;Allen et al., 2006). While both the MOD16 and SWAT ET use the Penman-Monteith equation, the methods for estimating the parameters of the equation are significantly different between them. For instance, the SWAT Penman-Monteith implementation requires wind speed data for the computation of the aerodynamic resistance, while the MOD16 Penman-Monteith variant does not require wind speed data but instead uses the Biome-BGC model (Thornton, 1998) to estimate the aerodynamic resistance. This study does not seek to evaluate the individual accuracy of any method, but rather to compare the ET results from the water balance-based hydrological models AWRA-L and SWAT and the energy balance-based model (MOD16) over a complex terrain catchment. Two different land cover products are used in

the SWAT model in this study (The Geoscience Australia and the MODIS land cover products). The rationale for
this is to analyse the effect of land cover on the ET modelling in SWAT and also the use of the MODIS land cover
allows for a direct comparison with the MOD16 which uses the same land cover product.  The results will be
compared temporally on catchment scale and spatio-temporally on sub-catchment scales to identify the effects of
input data and other drivers of ET estimation in the MOD16 and SWAT ET algorithms.

While the MODIS evapotranspiration has been widely studied and compared to other methods, this is much less
the case for SWAT ET (Table 1) and the AWRA-L. Moreover, a graduated spatial scale comparison of the SWAT
and MOD16 ET products is yet to be documented over a complex terrain. The objectives of this study are
therefore: (1) To simulate and compare the results of the evapotranspiration of SWAT, AWRA-L and MOD16
over a complex terrain at a catchment scale in a semi-arid climate; (2) To analyse and determine the spatial scale
at which the SWAT and MOD16 ET models tend towards agreement to enhance the confidence in ET estimation
in a complex terrain.

**Table 1:** Literature studies of MODIS and SWAT evapotranspiration (see Table 2 for climate classification)

| Study Type | Reference | Method | Climate | Land Cover Cover | Spatial & temporal extents |
|---|---|---|---|---|---|
| MOD16 vs micrometeorological methods | Ruhoff et al. (2013) | EC validation at 2 sites | Cwa, Cfa | Savanna | 3 km x 3 km area, 8 day |
| | Liu et al. (2013) | LAS validation at 3 sites | Dwa, Cwa | Orchards, Croplands | 1 km x 1 km, annual |
| | Mu et al. (2011) | EC validation at 46 site | Global | Global | Various |
| | Kim et al. (2012b) | EC validation at 17 sites | Af, Dfb, Dwa, Cfa, Bsk, Am, ET, Aw, Dwc, Dfc, Dfd | Forest, croplands, grassland | 3 km x 3 km area, 8 day, 2000-2006 |
| | Velpuri et al. (2013) | EC validation at 60 sites | Bsk, Cfa, Csa, Csb, Dfa, Dfb, Dfc | Cropland, Forest, Woody Savanna, Grassland, Shrubland, Urban | Point scale at EC sites across the United States of America, monthly, 2001 - 2007 |
| MOD16 vs energy balance models | Jia et al. (2012) | MOD16 validation of ETWatch system | Dwa, Cwa | Farmland, Forest, Grassland,Shrub Forest, Beach land, Bare land, Urban, Paddy field | (1 km x 1 km grid over 318,000 km$^2$ ), annual , 2002-2009 |
| | Velpuri et al. (2013) | MOD16 vs Gridded Fluxnet ET (GFET) | Bsk, Cfa, Csa, Csb, Dfa, Dfb, Dfc | Cropland, Forest, Woody Savanna, Grassland, Shrubland, Urban | 50km, monthly, over the entire United States of America |
| MOD16 vs hydrological models | Ruhoff et al. (2013) | MOD16 vs MGB-IPH model | Cwa, Cfa | Forest, Shrubland, Savanna, Woody Savanna, Grassland, Cropland, Urban, Barren land | (1 km x 1 km grid over 145,000 km$^2$ ), 8 day, 2001 |
| | Trambauer et al. (2014) | MOD16 vs GLEAM, ERAI, ERAL, PCR-GLOBWB, PCR-PM, PCR-TRMM, PCR-Irrig | Various | Various | 1km$^2$, 0.25º, 0.5º, and ~0.7º resolutions over most of the African continent, daily and monthly, 2000 -2010 |

| | Velpuri et al. (2013) | MOD16 vs Water Balance ET (WBET) | Bsk, Cfa, Csa, Csb, Dfa, Dfb, Dfc | Cropland, Forest, Woody Savanna, Grassland, Shrubland, Urban | (1 km x 1 km over the entire United States of America), Annual, 2002-2009, |
| --- | --- | --- | --- | --- | --- |
| SWAT vs energy balance models | Gao and Long (2008) | SWAT vs SEBS, SEBAL, P-TSEB, S-TSEB | Dwb | Woodland, Grassland, Cropland | 1850 km$^2$ , 23 June 2005 and 25 July 2005 ( 2 days only) |



**Table 2:** Köppen-Geiger Climate Classification system (Kottek et al., 2006)

| Main climate | Precipitation | Temperature |
| --- | --- | --- |
| **A – equatorial** | W – desert | h – hot arid |
| **B – arid** | S – steppe | k – cold arid |
| **C – warm temperate** | f – fully humid | a – hot summer |
| **D – snow** | s – summer dry | b – warm summer |
| **E – polar** | w – winter dry | c – cool summer |
| | m – monsoonal | d – extremely continental |
| | | F – polar frost |
| | | T – polar tundra |

e.g Cwa – Warm temperate, winter dry, hot summer
**2   Model Description**
**2.1   SWAT Model**
The Soil and Water Assessment Tool (SWAT) is a physically based, semi-distributed hydrological model
designed on the water balance concept. SWAT simulates catchment processes such as evapotranspiration, runoff,
crop growth, nutrient and sediment transport on basis of meteorological, soil, land cover data and operational land
management practices (Neitsch et al., 2011). The SWAT model has been used in hydrological modelling from
sub-catchment scales of under 1 km$^2$ (Govender and Everson, 2005) to sub-continental scales (Schuol et al., 2008).
The model discretises a catchment into sub-catchments and further into hydrological response units (HRU), which
represent unique combinations of land cover, soil type and slope. The discretisation method employed by SWAT
enables the model to simulate catchment processes in detail and to understand the response of unique HRU's on
hydrological processes. Evapotranspiration is simulated at the HRU scale. A comprehensive outline of ET
calculations in SWAT is included in Appendix A and Fig. 1 summarizes in a flowchart the SWAT ET algorithm.
Where PET is the potential evapotranspiration, $E_{can}$ is the evaporation from canopy surface, $E_t$ is the transpiration,
$E_{soil}$ is the evaporation from the soil and Revap is the amount of water transferred from the underlying shallow
aquifer to the unsaturated zone in response to water demand for evapotranspiration.

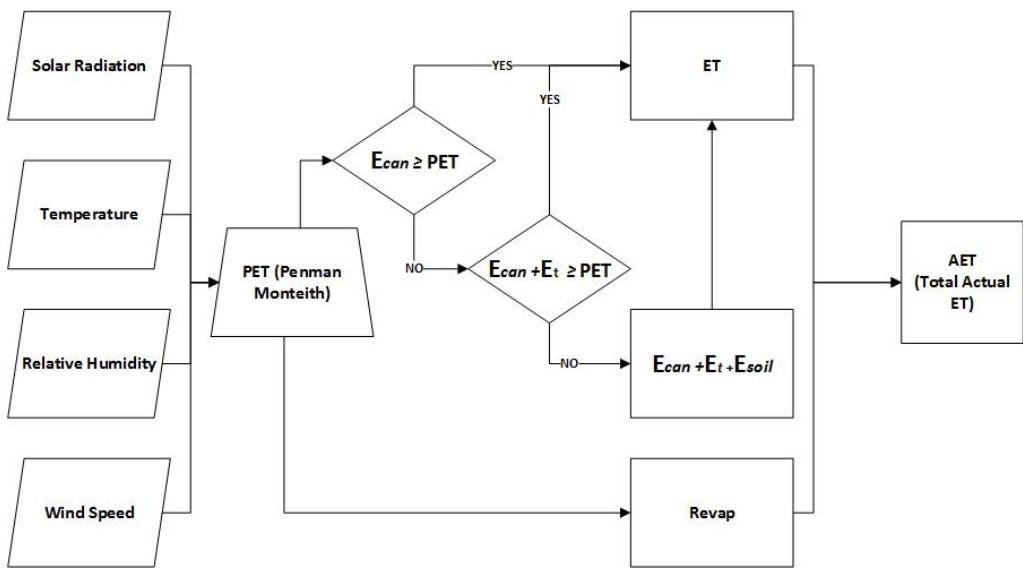


**Figure 1: SWAT ET flowchart (Penman-Monteith method)**

## 2.2    MOD16 Model
The MOD16 provides evapotranspiration estimates for $109.03 \times 10^6$ km$^2$ of global vegetated land area at 1 km$^2$
spatial resolution at 8 day, monthly and yearly temporal resolutions since the year 2000 (Mu et al., 2013). The
initial version of the MOD16 algorithm used MODIS imagery as part of a Penman-Monteith method as described
in Cleugh et al. (2007). The MOD16 algorithm was significantly improved by the inclusion of a sub-algorithm
for estimating soil evaporation as a component of total ET (Mu et al., 2007). Further improvements on the MOD16
algorithm such as the calculation and inclusion of night time evapotranspiration, partitioning of evaporation from
moist and wet soils were incorporated in the new algorithm (Mu et al., 2011). In this study, the ET products from
the new algorithm are used. Details of ET calculations in MOD16 are included in Appendix B while Fig. 2
summarizes in a flowchart the MOD16 ET algorithm.

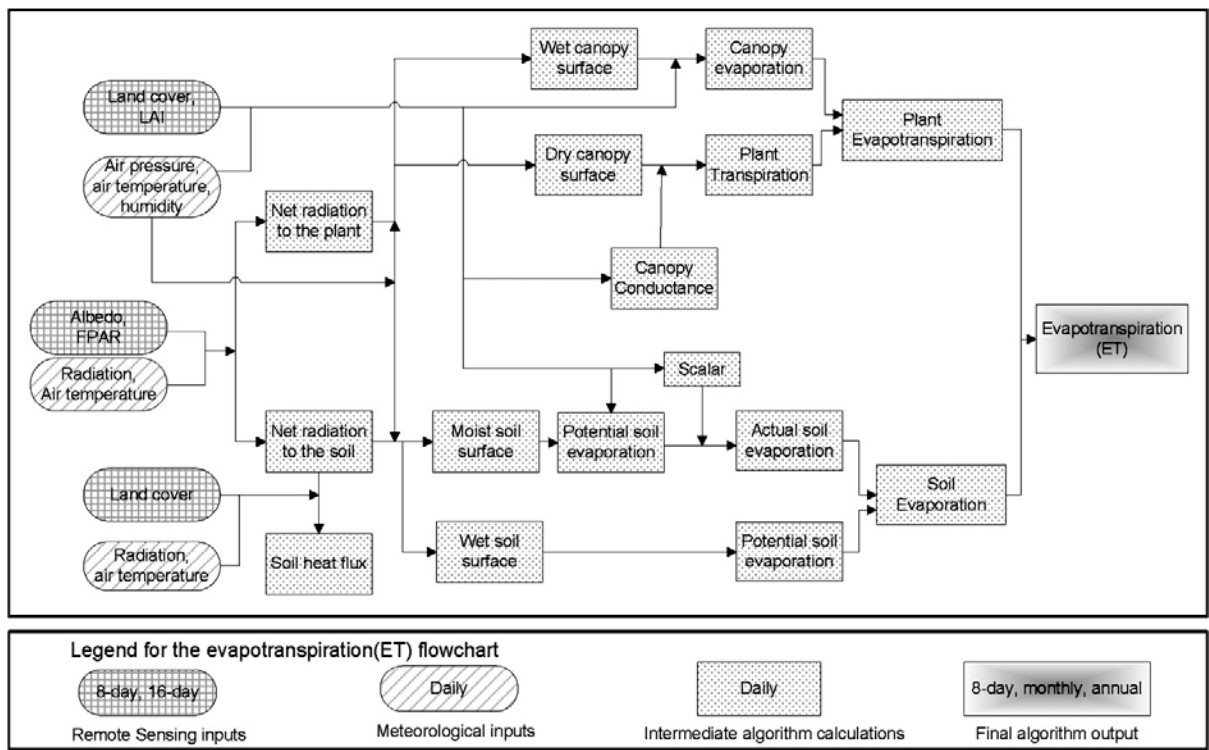


**Figure 2: Flowchart of the MOD16 ET algorithm (Mu et al., 2011)**

**2.3   AWRA-L Model**
The AWRA-L is a daily 25km$^2$ grid based hydrological model designed on the water balance concept over
Australia. The model conceptualises each grid as two distinct HRU's; shallow-rooted vegetation HRU and deep-
rooted vegetation HRU. The shallow-rooted vegetation corresponds to grass while the deep-rooted vegetation
corresponds to trees. The model conceptualises the soil into three layers with water storage capacity. The soil
surface storage with a 0.1m depth, the shallow storage from 0.1m to 1m and the deep storage from 1m to 6m.
The principal difference between the two HRU's is that the shallow-rooted vegetation HRU can only access the
first two soil storage layers while the deep-rooted vegetation HRU can access the 3 layers. The AWRA-L model
simulates catchment hydrological processes such as evapotranspiration, infiltration, runoff, drainage, interflow,
recharge amongst others.
Evapotranspiration in the AWRA-L is a sum of six processes; canopy evaporation from intercepted
precipitation, evaporation from soil surface, groundwater evaporation, shallow storage transpiration, deep
storage transpiration and groundwater transpiration. The evaporation in the model is constrained by the
Penmann equation (Penman, 1948). For a detailed structure of the AWRA-L model, see Viney et al. (2014).

**2.4   Penman-Monteith Algorithm Parameterization**
The MOD16 and SWAT ET algorithm, which are both based on the Penman-Monteith equation but parameterized
differently, suggests there will be similarities and differences in the results from both methods. Both algorithms
are principally limited on temporal timescales by the available energy to convert liquid water to atmospheric water
vapour. Their transpiration and soil evaporation algorithms are also very dependent on vegetation/biome type,
VPD, and the soil moisture constraint parameterization (Fig. 3).

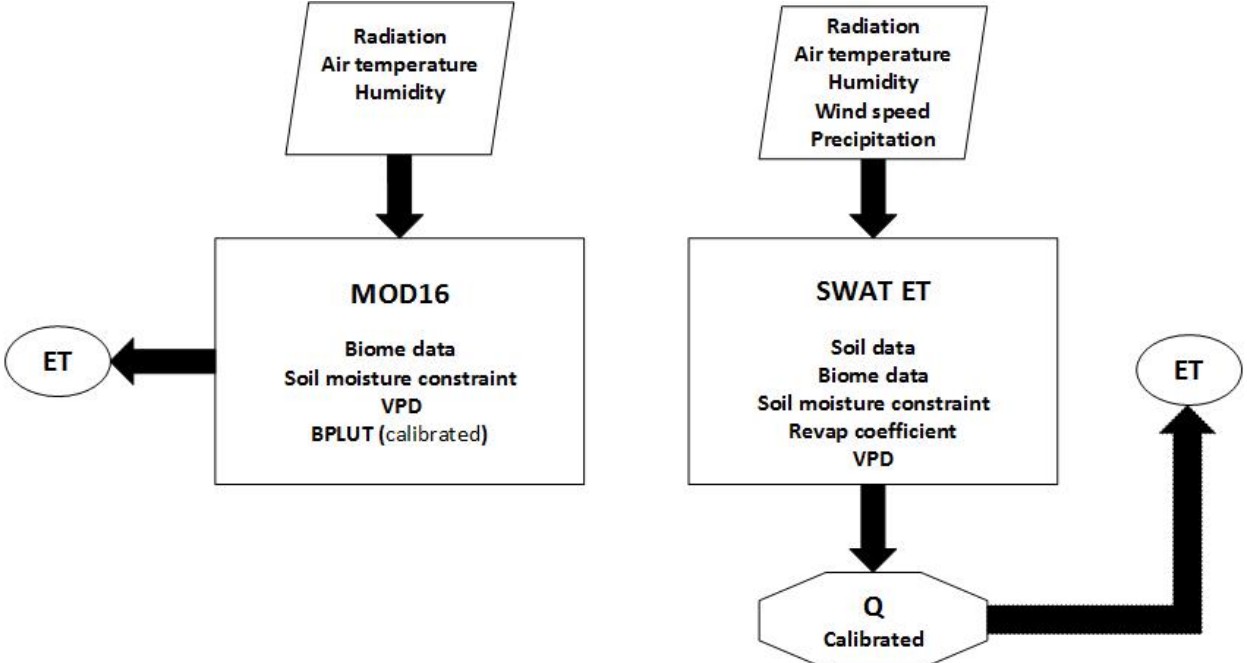


**Figure 3: MOD16 and SWAT ET parameterization (Q: discharge, BPLUT: biome properties lookup table; VPD:**
**vapour pressure deficit).**

In the SWAT ET algorithm, the VPD significantly impacts the transpiration through the constraining of the
stomatal conductance. Detailed soil data on HRU scale such as layer depth, number of layers, unsaturated
hydraulic conductivity and water capacity are crucial for constraining the soil moisture content, which in turn
regulates the percolation and recharge into the system. Similarly, the calculated MOD16 ET is significantly
impacted by the biome properties lookup table (BPLUT) and the soil moisture constraint function. The BPLUT
was calibrated using the response of biomes on flux tower sites globally. The BPLUT contains information on the
stomatal response of each biome to temperature, VPD and biophysical parameters. The soil moisture constraint
function is applied in the estimation of the soil evaporation and is an important parameter in regions where the
saturated zone is close to the ground surface such as our study area.


## 3   Data and Methods

### 3.1   Study Area

The study area is the Sixth Creek Catchment of South Australia, located in the western part of the Mount Lofty Ranges, which is a range of highlands separating the Adelaide Plains in the west from the Murray-Darling basin in the east. The western part of the Mount Lofty Ranges runs 90 km north to south, its summit is at 680 mAHD (metres Australian Height Datum) (Sinclair, 1980). It extends from the southernmost part at McLaren Vale on the Fleurieu Peninsula to Freeling in the north over an area of 2189 km$^2$. The Sixth Creek Catchment is a complex area, with acute elevation changes over few hundred metres (Fig. 4). The catchment is located close to the summit of the Western Mount Lofty Ranges.

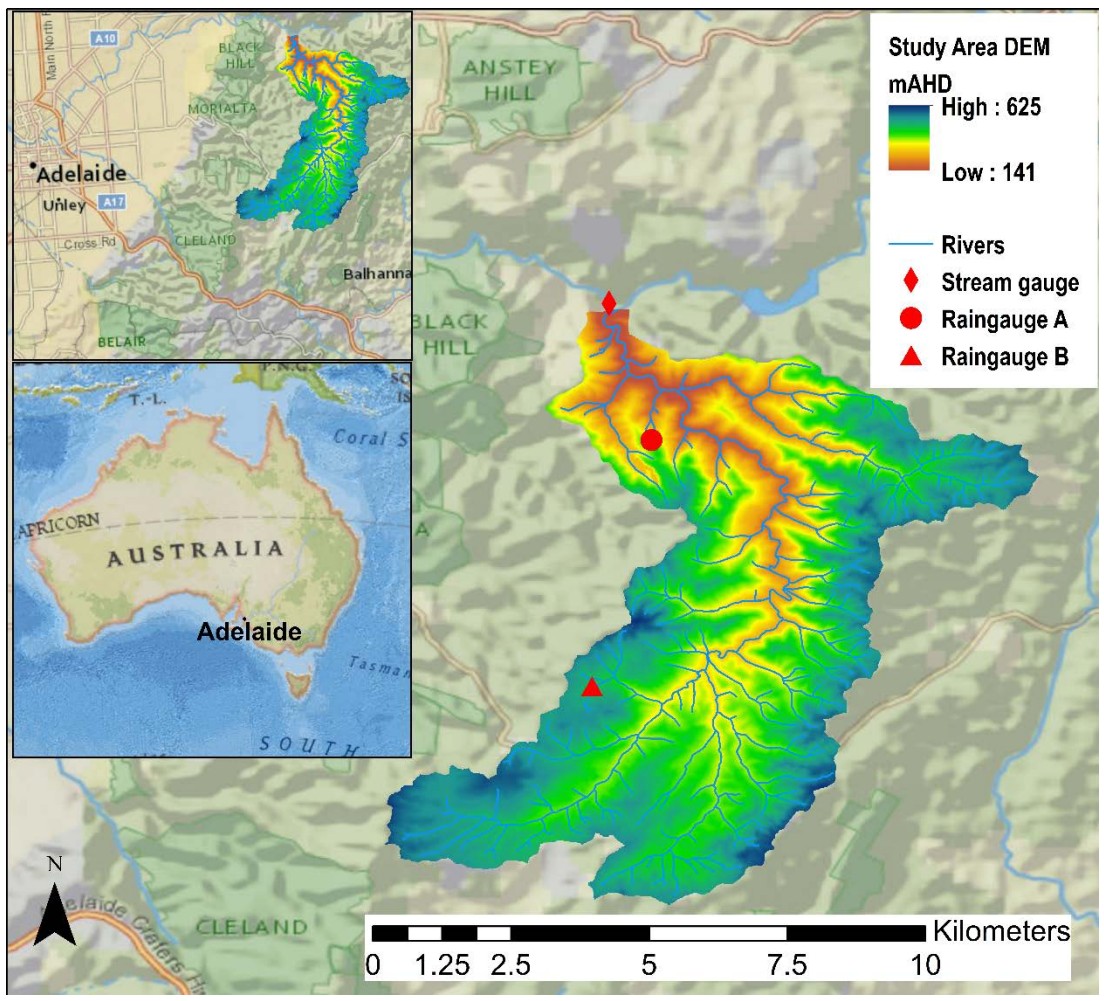

**Figure 4: Digital elevation model of the Sixth Creek Catchment study area (Gallant et al., 2011),**

It covers an area of 44 km$^2$ between $34°52′6.098″$ to $34°57′54.541″$S and $138°42′55.855″$ to $138°49′27.174″$E and has an elevation range of 140 - 625 mAHD (Fig. 4). The land cover consists of 95% forestland with significant deep-rooted Eucalyptus plantation and 5% pasture, shrubs and grasslands (Fig. 5b). Most of the native vegetation

is under conservation. The climate is Mediterranean, with warm dry summers and cool wet winters, and is of the
type "Csb" according to the Köppen-Geiger classification. The Sixth Creek is a perennial stream with mean annual
discharge of $0.25m^3$/s which accounts for 20 – 25 % of the mean annual rainfall in the catchment. The Sixth Creek
did however experience a total of 35 days of no flow in the 13-year period of this study (2000 – 2013) which
encompasses the "millennium drought years" (2000 – 2009) in Australia. The Sixth Creek is a gaining stream
with groundwater discharging into the stream and sustaining it especially during the dry summer months. The
depth to groundwater varies greatly across the complex terrain catchment, from less than 1 m to over 20 m across
the seasons.

The Sixth Creek Catchment's complex terrain plays a significant role in its hydrology, with highly localised
precipitation events recorded from the two weather stations in the catchment within the study period. The weather
stations are located 4.5 km apart with elevation difference of over 200 metres (Fig. 4). Differences in annual
rainfall of over 400 mm have been recorded between the two weather stations.
The annual precipitation for the period 2002 till 2016 for Station A ranges between 500 – 900 mm and 750-1500
mm for Station B, while the temperature ranges between 10.5 ℃ and 22.2 ℃ in the summer months and 3.4 ℃
and 10 ℃ in the winter months.

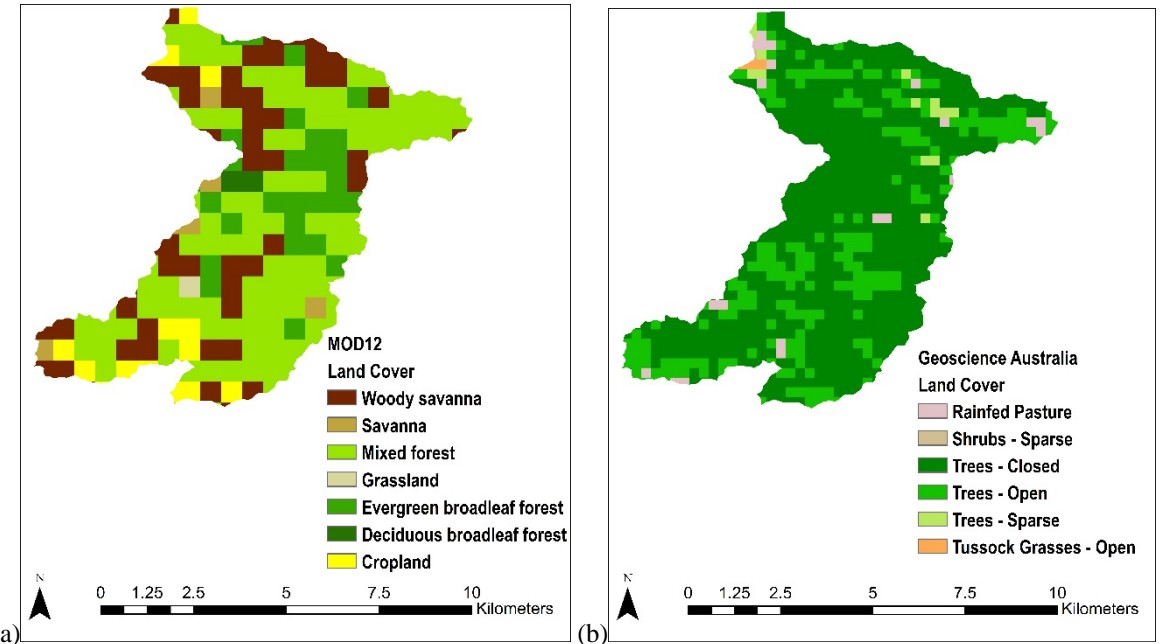

(a)    (b)
**Figure 5: (a) MOD12 land cover used in MOD16 (Friedl et al., 2010); (b) Geoscience Australia land cover**
**(Lymburner et al., 2010)**


**3.2 Input datasets**
The GIS interfaced version of SWAT (ArcSWAT) was used in the hydrological modelling. A 30 m Digital
Elevation Model (DEM) (Dowling et al., 2011) of the Sixth Creek Catchment was used to extract the stream
network and the catchment area. A detailed soil properties database for the catchment was created from the soil
data obtained from the Australian Soil Resource Information System (Johnston et al., 2003). The 250 m land cover
map of Australia from Geoscience Australia's Dynamic Land Cover database (Fig. 5b) is typically preferred to
be used in the SWAT model ahead of the 500 m MOD12 land cover map (Fig. 5a) due to its finer spatial resolution
and better biome match with local field knowledge but for direct comparison with MOD16, both maps are used
to run separate SWAT models. In this study, the $0.01^\circ \times 0.01^\circ$ wind speed data (McVicar et al., 2008), and the
$0.05^\circ \times 0.05^\circ$ relative humidity, temperature, rainfall, solar radiation (Jeffrey et al., 2001), were preferred to
weather station data. Four $0.05^\circ \times 0.05^\circ$ gridded data cells fall within the boundaries of the catchment and are
therefore comparable to the climate components of the two weather stations in the catchment. Moreover, the
gridded data used in this study are calibrated using the weather stations across Australia including the two weather
stations in the Sixth Creek Catchment, thus maintaining excellent correlation when compared to the weather
stations' measured data. Details of the gridded data methodology and algorithm used in this study can be found
in Jeffrey et al. (2001) and McVicar et al. (2008). The daily gridded climate datasets were simply averaged over
the Sixth Creek Catchment, to obtain values used in this study.

The monthly MOD16 datasets for the years 2000 to 2013, at 1 km$^2$ spatial resolution were used in this study (Mu
et al., 2013). Catchment averages were calculated by simple averaging of all the 1 km$^2$ cells that fall within the
catchment area.

**3.3 SWAT Model Setup and Calibration**
The soil, land cover and DEM derived slope data were classified into classes and used to create 124 and 119
unique HRU's for the Geoscience Australia and MOD12 land covers respectively, ranging from 0.001 km$^2$ to 6
km$^2$ in area. While each unique HRU has specific set of properties several small areas with the same land cover,
slope and soil type make up the total area of a single HRU. The properties of each unique HRU determine how it
responds to precipitation, and how different hydrological processes such as streamflow, runoff, lateral flow and
evapotranspiration are modelled in the catchment. The runoff from each HRU is accumulated and routed through
the river network to the outlet of the catchment. Driven by the meteorological input, the model simulates
catchment hydrological processes with a daily time step for the period 2000 to 2013.

The SWAT model is calibrated by fitting simulated streamflow to observed streamflow with the SUFI-2
algorithm. This semi-automatic Latin hypercube sampling algorithm optimizes SWAT model parameters while
attempting to fit the simulated data as close as possible to the observed data using the user preferred objective
function from those detailed below as measurement of simulation accuracy (Abbaspour, 2007). Although a single
user objective function is used in the calibration and validation, the results of the other objective functions are
also recorded for the optimal model run.

Nash Sutcliffe Efficiency ($N_{SE}$) (Nash and Sutcliffe, 1970),
$$N_{SE} = 1 - \frac{\sum_{n=1}^{N}(Q_n - \widehat{Q_n})^2}{\sum_{n=1}^{N}(Q_n - \overline{Q})^2} \tag{1}$$
where $Q_n$ (m$^3$s$^{-1}$) is the measured discharge at time $n$, $\widehat{Q_n}$ (m$^3$s$^{-1}$) is the simulated discharge at time $n$, $\overline{Q}$ (m$^3$s$^{-1}$)
is the mean measured discharge and $N$ is the number of time steps.

Ratio of root mean squared error to the standard deviation of measured data ($R_{SR}$) (Moriasi et al., 2007),
$$R_{SR} = \frac{\sqrt{\sum_{n=1}^{N}(Q_n - \widehat{Q_n})^2}}{\sqrt{\sum_{n=1}^{N}(Q_n - \overline{Q})^2}} \tag{2}$$

Percent bias ($P_{BIAS}$),
$$P_{BIAS} = 100 \frac{\sum_{n=1}^{N}(Q_n - \widehat{Q_n})}{\sum_{n=1}^{N} Q_n} \tag{3}$$

Coefficient of determination (R$^2$),
$$R^2 = \left( \frac{(\sum_{n=1}^{N}(Q_n - \overline{Q})(\widehat{Q_n} - \widetilde{Q_n}))}{\sqrt{\sum_{n=1}^{N}(Q_n - \overline{Q})^2}\sqrt{\sum_{n=1}^{N}(\widehat{Q_n} - \widetilde{Q_n})^2}} \right)^2 \tag{4}$$
where $\widetilde{Q_n}$ (m$^3$s$^{-1}$) is the mean simulated discharge.

Kling-Gupta Efficiency ($K_{GE}$) (Gupta et al., 2009),
$$K_{GE} = 1 - \sqrt{(r-1)^2 + (\alpha - 1)^2 + (\omega - 1)^2} \tag{5}$$
where $r$ is the linearcorrelation coefficient between the simulated and measured variable, $\omega = \frac{\widetilde{Q_n}}{\overline{Q}}$ , $\alpha = \frac{\sigma_s}{\sigma_m}$, $\sigma_s$
and $\sigma_m$ are the standard deviation of simulated and measured data.
After obtaining a satisfactory fit between the simulated and observed streamflow data during calibration, the
model is validated by running the model for a different time period using the same parameters from the calibration
period. SUFI-2 further incorporates the unitless P and R-factor metric, which gives an indication of the confidence
in the calibration exercise. The P-factor which is also referred to as the 95 Percent Prediction Uncertainty (95PPU),
is the percentage fraction of observed data captured which falls between the 2.5 and 97.5 percentiles, while the
R-factor is the width of the 95PPU. The P and R-factors are iteratively determined using Latin Hypercube
Sampling. For streamflow calibration and validation to be considered reliable, combined satisfactory values
should be obtained of P-factor ($> 0.7$), R-factor ($< 1$) (Abbaspour, 2007) and of one of the objective functions,
$N_{SE}$ ($> 0.5$), $R_{SR}$ ($\leq 0.7$) and $P_{BIAS}$ ($\pm 25\%$) (Moriasi et al., 2007). In this study, the NSE objective function
combined with the P and R factors are used. The result of the other objective functions at the optimal NSE are
also recorded. For a comprehensive explanation of the SUFI-2 algorithm, see Abbaspour (2007).
The calibration process was conducted on daily timescales for the years 2000 to 2005 while the validation was
conducted for the years 2007 to 2013. A warm up period of 5 years between 1995 and 1999 was used in the SWAT
model to equilibrate the model mass budget and internal reservoirs. The relatively long periods of streamflow
calibration and validation on daily timescales were specifically used to address the potential problem of
equifinality of parameters to be optimized. The principle of equifinality has been known to affect semi-distributed
models such as SWAT (Qiao et al., 2013). Nevertheless, the use of many observation points has been observed to
effectively constrain it (Tobin and Bennett, 2017). In this study, 21 sensitive SWAT model parameters (Table 3)
are optimized with SUFI-2 to fit simulated streamflow to the observed streamflow data. In the SUFI-2 algorithm
preparation for calibration, an "r_" and a "v_" prefix before a SWAT model parameter (Table 3) are indicative of
a relative change (a percentage increase or decrease in the SWAT modelled value) and replacement change of the
original SWAT modelled values respectively. The relative change is often used to fine tune parameters that have
been modelled within the acceptable range while the replacement change is used when modelled parameter values
are at odds with local field knowledge or established values.
The resultant SWAT simulated ET was compared with the MOD16 ET using the root mean square error ($R_{MSE}$),
mean difference ($M_D$), Pearson's correlation coefficient (R) and coefficient of determination ($R^2$) metrics.
$$R_{MSE} = \sqrt{\frac{\sum_{n=1}^{N}(x_{1,n}-y_{1,n})^2}{N}}$$ (6)
Where $x_1$ and $y_1$ are SWAT and MOD16 monthly ET values respectively.
$$M_D = \left(\frac{x_1+x_2...x_N}{N}\right) - \left(\frac{y_1+y_2...y_N}{N}\right)$$ (7)
$$R = \frac{(\sum_{n=1}^{N}(Q_n-\overline{Q})(\widehat{Q_n}-\widetilde{Q_n}))}{\sqrt{\sum_{n=1}^{N}(Q_n-\overline{Q})^2}\sqrt{\sum_{n=1}^{N}(\widehat{Q_n}-\widetilde{Q_n})^2}}$$ (8)

**Table 3:** Optimized SWAT parameters and their final range

| P | Parameter Description | Final Parameter Range |
|---|---|---|
| r _ | SCS Runoff Curve Number for moisture condition II | $[1 + (-0.048 - 0.122)] \times Actual\ value$ |
| v | Baseflow recession constant (days) | $0.58 - 0.93$ |
| v | Groundwater delay time (days) | $1.89 - 3.70$ |
| v | Groundwater "Revap" coefficient | $0.12 - 0.2$ |
| v | Soil evaporation compensation factor | $0.2 - 0.5$ |
| v _ | Manning's "n" value for the main channel | $0.05 - 0.15$ |
| r | Surface runoff lag coefficient | $[1 + (0.22 - 1.2)] \times Actual\ Value$ |
| v _ | Baseflow alpha factor for bank storage (days) | $0.5 - 1$ |
| v _ | Available water capacity of the soil layer (mm/mm) | $0.24 - 0.71$ |
| r _ | Saturated hydraulic conductivity (mm/hr) | $[1 + (-0.99 - -0.39)] \times Actual\ Value$ |
| r | Moist bulk density (g/cm$^3$) | $[1 + (-0.37 - -0.04)] \times Actual\ Value$ |
| r _ | Depth from soil surface to bottom of layer (mm) | $[1 + (-0.25 - -0.04)] \times Actual\ Value$ |
| v | Plant uptake compensation factor | $0.77 - 1$ |
| v _ | Threshold depth of water in the shallow aquifer required for return flow to occur (mm) | $0 - 500$ |

| | | | |
|---|---|---|---|
| **v** | Initial depth of water in the shallow aquifer (mm) | | $20000 - 30000$ |
| _ | | | |
| **v** | Initial depth of water in the deep aquifer (mm) | | $10000 - 20000$ |
| _ | | | |
| **r** | Average slope steepness (m/m) | | $[1 + (-0.24 - 0.15)] \times Actual\,Value$ |
| **r** | Manning's "n" value for overland flow | | $[1 + (-0.84 - -0.05)] \times Actual\,Value$ |
| **r** | Average slope length (m) | | $[1 + (-0.9 - -0.24)] \times Actual\,Value$ |
| **v** | Threshold depth of water in the shallow aquifer required for Revap to occur (mm) | | $0 - 100$ |
| **v** | Effective hydraulic conductivity in main channel alluvium (mm/hr) | | $6 - 30$ |
| _ | | | |


## 4  Results
### 4.1  Streamflow

The streamflow was calibrated and validated on daily timescales according to the guidelines set out in Moriasi et

al. (2007) and Abbaspour (2007) (Table 4, Fig. 6). The result indicates an observed data bracketing of between

87% and 89% for both calibration and validation with R-factors under 1.

**Table 4:** Streamflow calibration and validation results

| Model | | P-factor | R-factor | $N_{SE}$ | $R^2$ | $K_{GE}$ | $R_{SR}$ | $P_{BIAS}$ |
|---|---|---|---|---|---|---|---|---|
| SWAT with Geoscience Land Cover | Calibration | 0.89 | 0.66 | 0.61 | 0.62 | 0.71 | 0.62 | -11.1 |
| | Validation | 0.87 | 0.91 | 0.78 | 0.78 | 0.88 | 0.47 | -0.1 |
| SWAT with MOD12 Land Cover | Calibration | 0.88 | 0.69 | 0.62 | 0.64 | 0.74 | 0.61 | -13.5 |
| | Validation | 0.87 | 0.98 | 0.79 | 0.80 | 0.87 | 0.46 | -6.5 |

312

Table 4 shows better results for the validation than calibration for the $N_{SE}$, $R^2$, $K_{GE}$ and $R_{SR}$ metrics, however

slightly lower for the P-factors. The results of the calibration and validation exercise on daily timescales show

that the model effectively represents the high and low flow periods (Fig. 6).

316

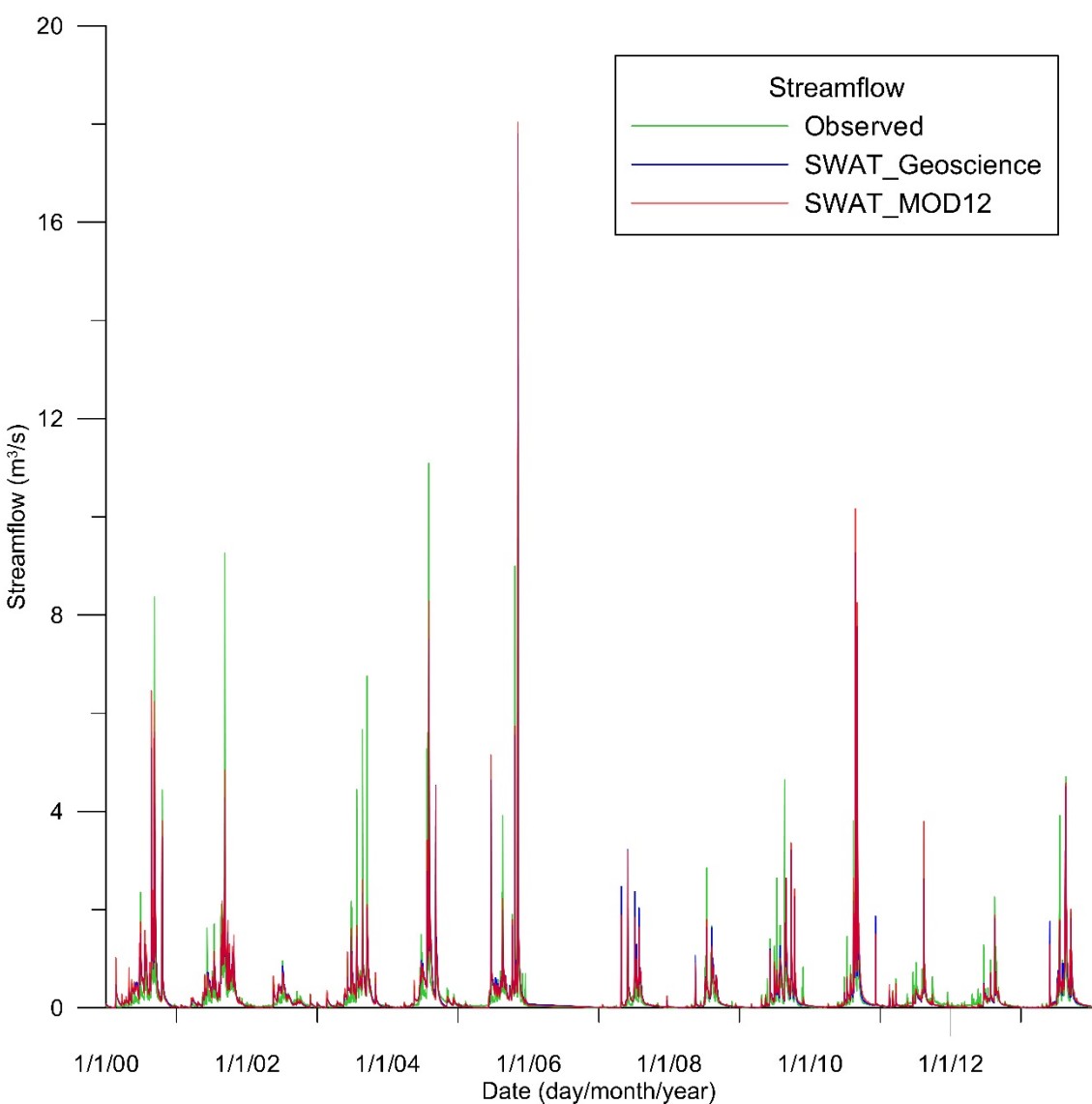

317

**Figure 6: Streamflow calibration (2000-2005) and validation (2007-2013)**

**4.2 Sub-catchment scale evapotranspiration**

The SWAT ET model is calculated at the HRU scale (Fig. 7a & 7b), however for direct comparison with the

MOD16 ET (Fig. 7c), the HRU ET results were reprocessed into 1 km² cells using simple averaging. For cells on

the boundary which do not aggregate up to the 1km² resolution, a percentage weighting based on the area covered

is applied. Figure 7d shows the mean annual difference between both SWAT models (the SWAT model with

Geoscience land cover as SWATGEO and the SWAT model with MOD12 land cover as SWATMOD12) over

the validation period at the 1 km² spatial resolution. The SWATMOD12 and the MOD16 maps (Fig. 7b and 7c)

can be seen to show some spatial semblance in the north, south, east and west corners of the catchment principally

due to the use of the MOD12 map in both models. Generally, a trend of higher ET in the north-east and central

part of the catchment is seen while lower ET is observed in the south-western parts of the catchment. The spatially

distributed mean annual ET difference of the SWAT models compared to the MOD16 show about 40% of the

catchment with a difference of ±100 mm/year at the 1 km$^2$ spatial scale. Clear spatial difference between the
SWAT models are seen at the HRU scale but at the 1 km$^2$ resolution, the maximum mean annual difference
between the SWAT models is 12%.

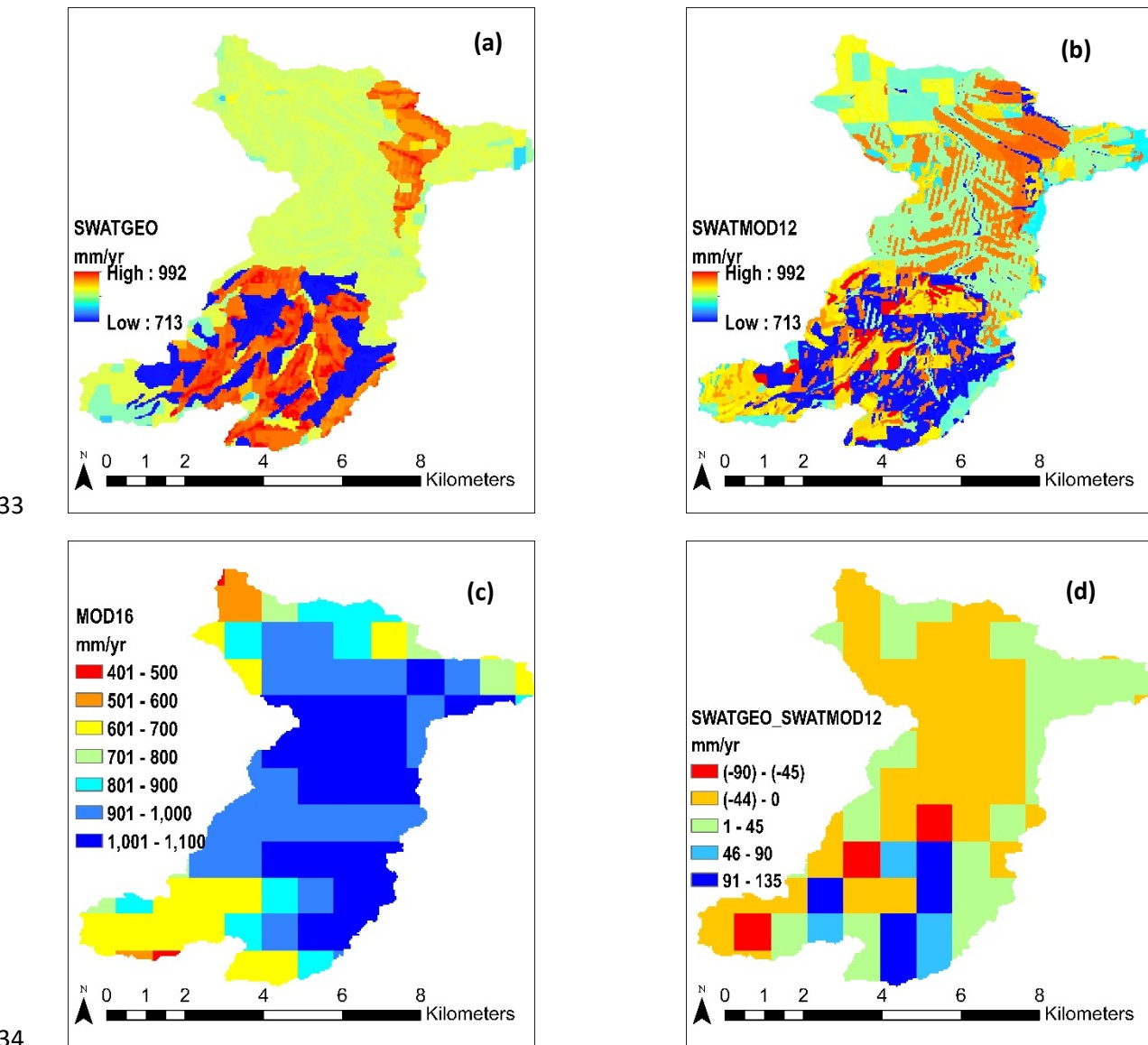



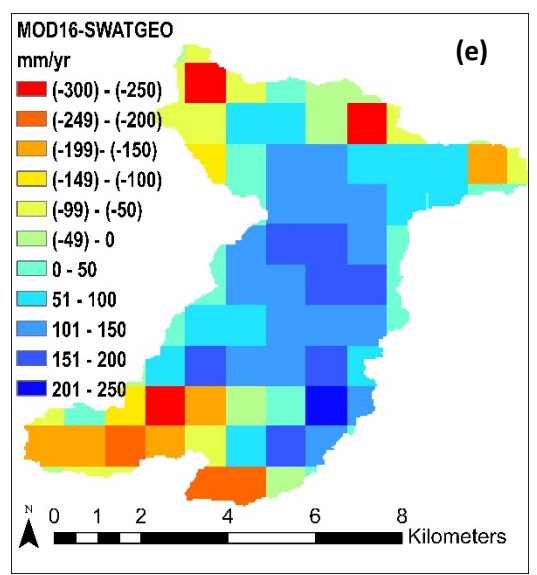 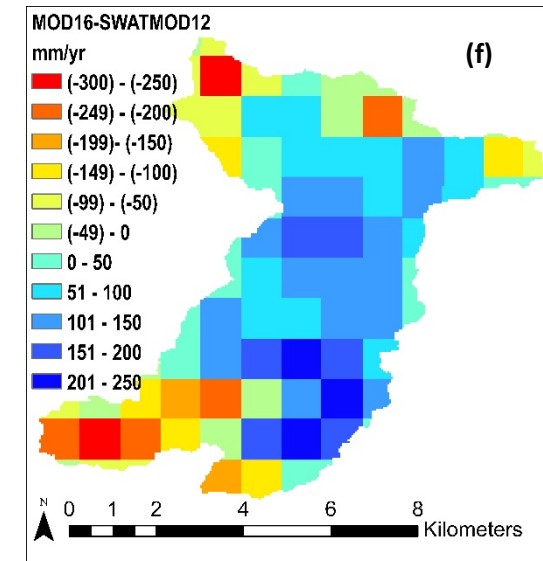

**Figure 7: (a) HRU scale SWATGEO mean ET (2007-2013); (b) HRU scale SWATMOD12 mean ET (2007-2013; (c) 1 km² grid MOD16 mean ET (2007-2013); (d) Mean difference between SWATGEO and SWATMOD12 for corresponding 1 km² grid cells (2007-2013); e) Mean difference between MOD16 and SWATGEO for corresponding 1 km² grid cells (2007-2013); (f) Mean difference between MOD16 and SWATMOD12 for corresponding 1 km² grid cells (2007-2013)**

Further analyses were carried out to determine the effect of spatial aggregation on the correspondence between the ET methods. For the spatial aggregation analysis, the SWATGEO model was used due to its improved land cover accuracy based on field knowledge. The box and whisker plot in Fig. 8 shows the spread of the difference between the SWAT ET and the MOD16, with the bottom, middle and top of the box indicating the 25[th], 50[th] and 75[th] quartiles of the distribution. The lowest and highest bars in the plot indicate the minimum and maximum differences between the ET products at the different spatial scales. Figure 8 show that with increasing cell aggregation the difference in the ET between SWAT and MOD16 decreases. At 1 km², 4 km², 9 km², 16 km² and 25 km² the maximum cell difference between the SWAT and MOD16 ET are 31%, 19%, 15%, 11% and 9% respectively.

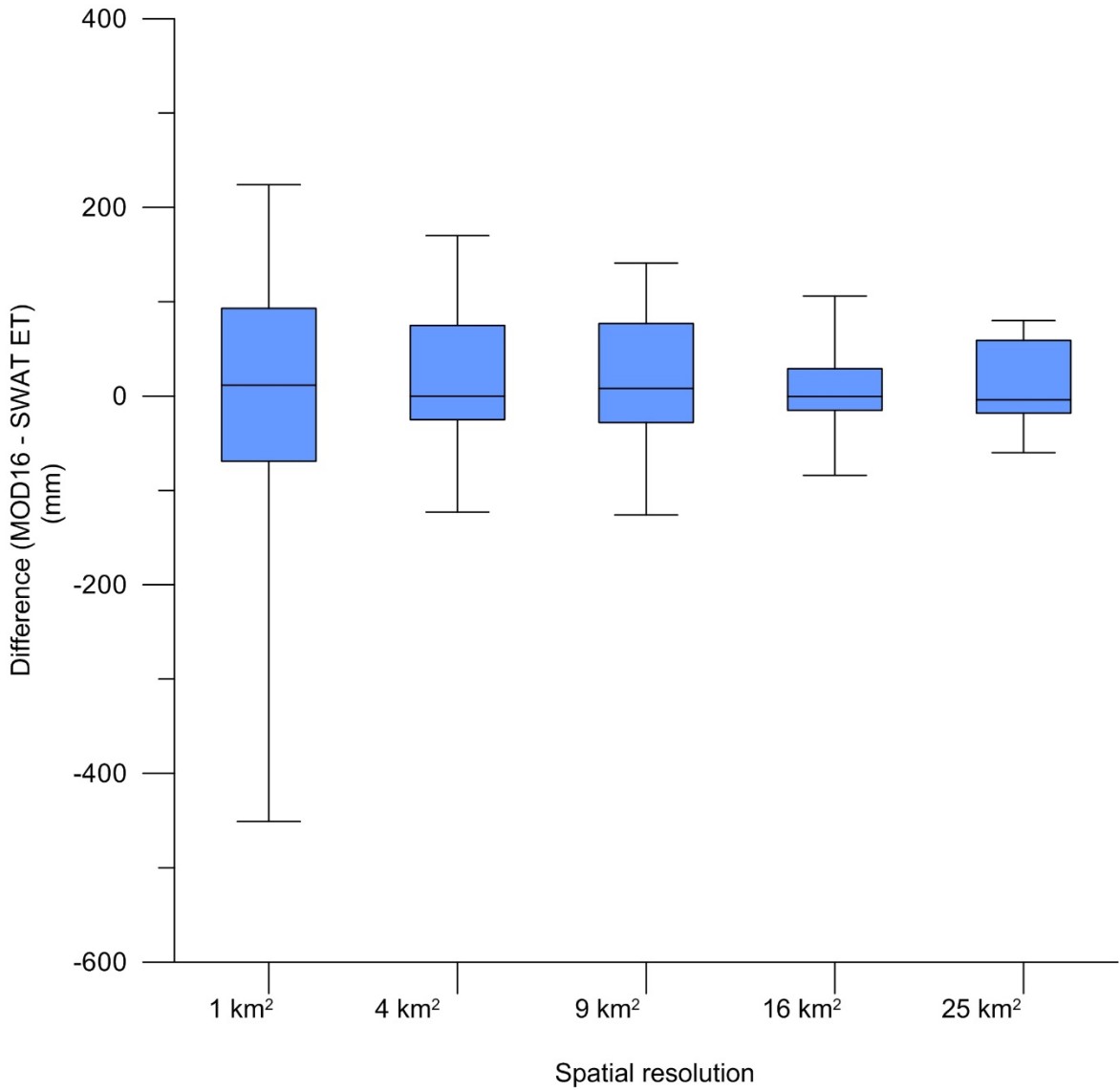


**Figure 8: Differences between SWATGEO ET and MOD16 for spatial aggregations between 1 and 25 km². The**
**bottom, middle and top of the whisker indicate the 25th, 50th and 75th quartiles of the distribution, the lowest and**
**highest bars indicate the minimum and maximum differences.**



The grand variances for the monthly data of the three models were calculated and partitioned into the spatial and
temporal components at the 1 km², 4 km², 9 km², 16 km² and 25 km² resolutions (Table 5) using the Time-First
formulation described in Sun et al. (2010). The partitioning presents the average of the temporal variances for
each of the regions in the catchment as the temporal component and the spatial variance of the
evapotranspiration as the spatial component shows the spatial component consistently higher across the three
models. The partitioning shows that at the finer resolution the variance in the evapotranspiration in the models
are principally associated with the spatial component but the temporal component of the variance increases with
spatial aggregation.




Table 5: Variance partitioning into space and time components at various spatial resolutions

| Spatial Resolution | Model | Spatial Component in mm$^2$ (%) | Temporal Component in mm$^2$ (%) |
|---|---|---|---|
| 1 km$^2$ | SWATMOD12 | 74.4 (80.9) | 17.6 (19.1) |
| | SWATGEO | 75.5 (80.6) | 18.2 (19.4) |
| | MOD16 | 82.5 (84.9) | 14.7 (15.1) |
| 4 km$^2$ | SWATMOD12 | 239.9 (79.8) | 60.6 (20.2) |
| | SWATGEO | 241.1 (79.4) | 62.72 (20.6) |
| | MOD16 | 265.0 (84.04) | 50.34 (16.0) |
| 9 km$^2$ | SWATMOD12 | 434.4 (77.7) | 124.9 (22.3) |
| | SWATGEO | 434.8 (77.2) | 128.4 (22.8) |
| | MOD16 | 479.2 (82.0) | 105.1 (18.0) |
| 16 km$^2$ | SWATMOD12 | 586.2 (74.8) | 198.0 (25.2) |
| | SWATGEO | 590.7 (74.3) | 204.8 (25.7) |
| | MOD16 | 637.3 (80) | 159.4 (20) |
| 25 km$^2$ | SWATMOD12 | 665.9 (68.3) | 308.7 (31.7) |
| | SWATGEO | 669.9 (67.6) | 320.6 (32.4) |
| | MOD16 | 738.8 (73.5) | 266.4 (26.5) |




### 4.3   Catchment Scale Evapotranspiration


At catchment scale, the mean annual ET of the SWATGEO, SWATMOD12 and the MOD16 models are 873, 864
and 865mm respectively.  The means show better agreement between the SWATMOD12 and MOD16 models
which is attributed to the use of the same land cover in both models.
To compare the temporal dynamics of the MOD16, the SWAT ET and the AWRA-L ET, the data were aggregated
to catchment scale. As both SWAT models tend towards unity at the catchment scale with less than 1% difference
in their annual mean ET, only the SWATGEO model is evaluated at catchment scale as the more accurate model
to keep with the philosophy of the study.
Monthly MOD16 ET and AWRA-L ET values at 1 $km^2$ and 25 $km^2$ resolution respectively were averaged to
catchment scale values using the spatial analyst tools in ArcGIS, while ET values from the validated SWAT model
on catchment spatial extent and daily timescales were aggregated to monthly timescales. Using the $R_{MSE}$  and $R^2$
metrics the analysis shows a good correspondence between the models (Fig. 9). The SWAT and MOD16 methods
at catchment scale has a maximum annual ET difference and mean ET difference of respectively less than 13 and
6 percent for the period from 2007 to 2013.The MOD16 and the AWRA-L show similar temporal patterns, but
the AWRA-L ET was significantly lower than both the MOD16 and SWAT ET results (Fig. 9). A direct
comparison between the AWRA-L ET and the SWAT ET without the Revap component shows very high
correlation and agreement between both models with maximum annual ET difference and mean ET difference of
respectively 10 and 2 percent for the period from 2007 to 2013.

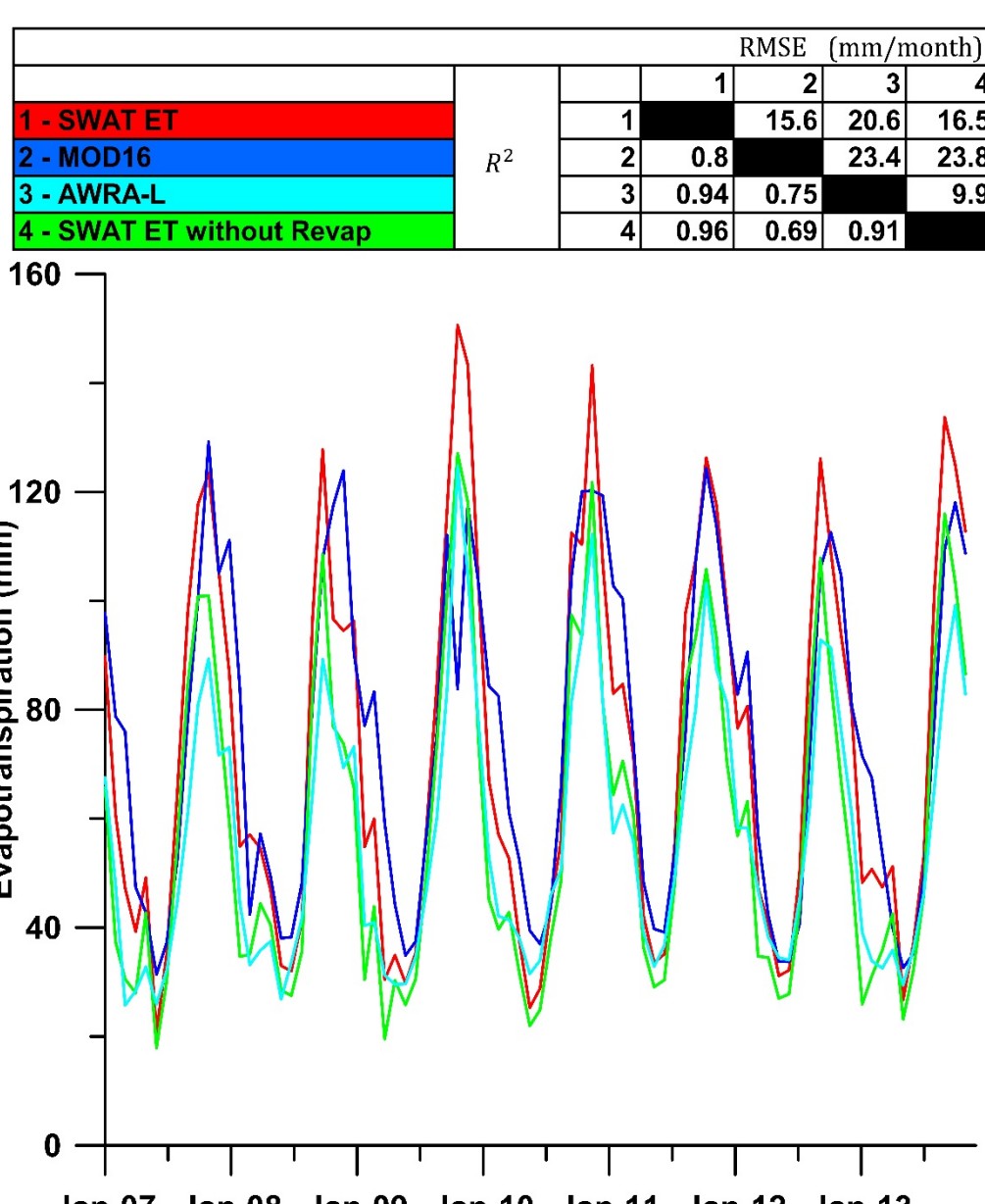

| | | RMSE | (mm/month) | | |
|---|---|---|---|---|---|
| | | | 1 | 2 | 3 | 4 |
| 1 - SWAT ET | 1 | | | 15.6 | 20.6 | 16.5 |
| 2 - MOD16 | $R^2$ 2 | | 0.8 | | 23.4 | 23.8 |
| 3 - AWRA-L | 3 | | 0.94 | 0.75 | | 9.9 |
| 4 - SWAT ET without Revap | 4 | | 0.96 | 0.69 | 0.91 | |

**Figure 9: Monthly Comparison of SWAT, AWRA-L and MOD16 ET at catchment scale.**

## 5   Discussion

### 5.1   Spatial Aggregation Analysis

The mean annual graduated spatial scale analysis across the SWAT models and the MOD16 for 2007-2013 exhibits a wide spread at the 1km$^2$ spatial resolution with a maximum cell difference of 31%. When the data was aggregated to 4 km$^2$ using the simple averaging method, the maximum difference reduced to an acceptable 19%.

Further aggregation to 9 km$^2$ reduced the maximum difference by a further 4% but also sees a significant
degradation in the resolution of the evapotranspiration data. Table 5 also shows the impact of the spatial
aggregation on the variance of the monthly ET data across the SWAT and MOD16 models. It is observed that the
aggregation from 1 km$^2$ to 4 km$^2$ altered the percentage variance between the spatial and temporal by about 1%
across the three models but beyond the 4 km$^2$ resolution the spatial component of the variance which accounts for
the larger portion of the variance begins to degrade further. Hence our spatial scale of confidence for small
catchment scale ET analysis is the 4 km$^2$ resolution based on the comparison of the SWAT and MOD16 ET over
a complex terrain.
The differences between regions in the catchment are more significant at finer spatial resolutions due to the diverse
input data and their associated errors, these impacts become less significant as the outputs are up-scaled (Fig. 8).
This trend was also observed by Hong et al. (2009). The simple averaging method was preferred in this study over
the bilinear, cubic and other methods as the simple averaging method has been observed to be the best in flux
aggregation after a study of various methods (Ershadi et al., 2013).
**5.2   Sources of differences across the three models**
The recognized principal sources of differences between the three ET methods are associated with land cover, the
Revap component in SWAT and the HRU parameterization in the AWRA-L; they are discussed in the following
sections.
**5.2.1   Land Cover**
The land cover is an important parameter in the MOD16 and SWAT ET algorithms as it determines the values
allocated to biophysical properties such as leaf conductance and boundary layer resistance, which significantly
impact ET calculations. The impact of the land cover on the SWAT models is evident from the spatially divergent
high-resolution SWAT models (Fig. 9a and 9b), at the HRU scale, though the streamflow calibration and
validation parameters and results were similar. With the spatial aggregation of the SWAT models to 1 km$^2$
resolution, the obvious spatial differences at the HRU scale reduces significantly and begins to disappear beyond
the 1 km$^2$ resolution. Differences in the land cover in the SWAT models were responsible for the difference spatial
distribution of the ET across the catchment between the models. The effect of the land cover on the MOD16 was
not evaluated, however, the SWATMOD12 model with the same land cover expectedly showed better agreement
when compared with the MOD16 with mean for the period of 2007-2013 within 1mm at the catchment scale. The
Geoscience Land cover map has 95% percent forests, while the MOD12 has a classification of 67% forests and
24 % woody savanna, with most of the region misclassified as woody savanna having some similar properties of
the forests. At catchment scale, the data averaging contributes to the convergence of the MOD16 and SWAT ET
results albeit with closer agreement between the MOD16 and SWATMOD12 which share land cover.

**5.2.2 Revap**
The Revap component of the AET in SWAT is mostly significant in forested catchments with deep rooted trees
that can access the saturated zone and as such are governed by land use parameters (Neitsch et al., 2011). However,
the relative accuracy of the Revap component of the ET on HRU scales has been questioned (Liu et al., 2015) due
to the linear relationship between the Revap coefficient and potential evapotranspiration in SWAT (see Eqn. A23).
The Revap component in this study appears consistent with the studies by Benyon et al. (2006) in south-eastern
Australia with similar climatic condition as the Sixth Creek Catchment. Benyon et al. (2006) observed that under
the combined conditions of highly permeable soils, available groundwater resources of low salinity (<2000 mg/L),
a high transmissivity aquifer and groundwater of depths up to 6 m, annual groundwater ET contribution to total
ET ranged from 13 – 72% for sampled Eucalyptus tree species. The Sixth Creek Catchment is principally
underlain by the highly transmissive and permeable Aldgate Sandstone aquifer, with salinity levels well below
2000 mg/L (Gerges, 1999). Monitoring bores in the Sixth Creek Catchment have recorded standing water levels
of less than 1.5 metres at the end of the rainy winter months in parts of the catchment. The Sixth Creek Catchment
has been identified as one of the principal recharge zones in the Western Mount Lofty Ranges based on the
catchment geology and hydrochemical analysis (Green and Zulfic, 2008). A significant portion of the 95%
forested part of the Sixth Creek Catchment is a mosaic of various Eucalyptus tree species, thereby corroborating
the results of Benyon et al. (2006). The AWRA-L ET model does not appear to include a separate groundwater
ET model in its algorithm such as is found in the SWAT model (A23-26), hence the correlation and strong
agreement between the AWRA-L model when the Revap is unaccounted for in the SWAT ET. The results suggest
the Revap is a significant contributor to ET in the Sixth Creek Catchment (Fig. 10) with mean annual contribution
of 20% for the years 2007 – 2013, while monthly contributions ranged from 15 – 52 % over the same period. The
possibility exists that the linear relationship with PET employed in its calculation on HRU scale may be
contributory to the higher range of ET fluctuation seen in the SWAT model on the 1 km$^2$ scale when compared to
the MOD16, however, that is beyond the scope of this study.

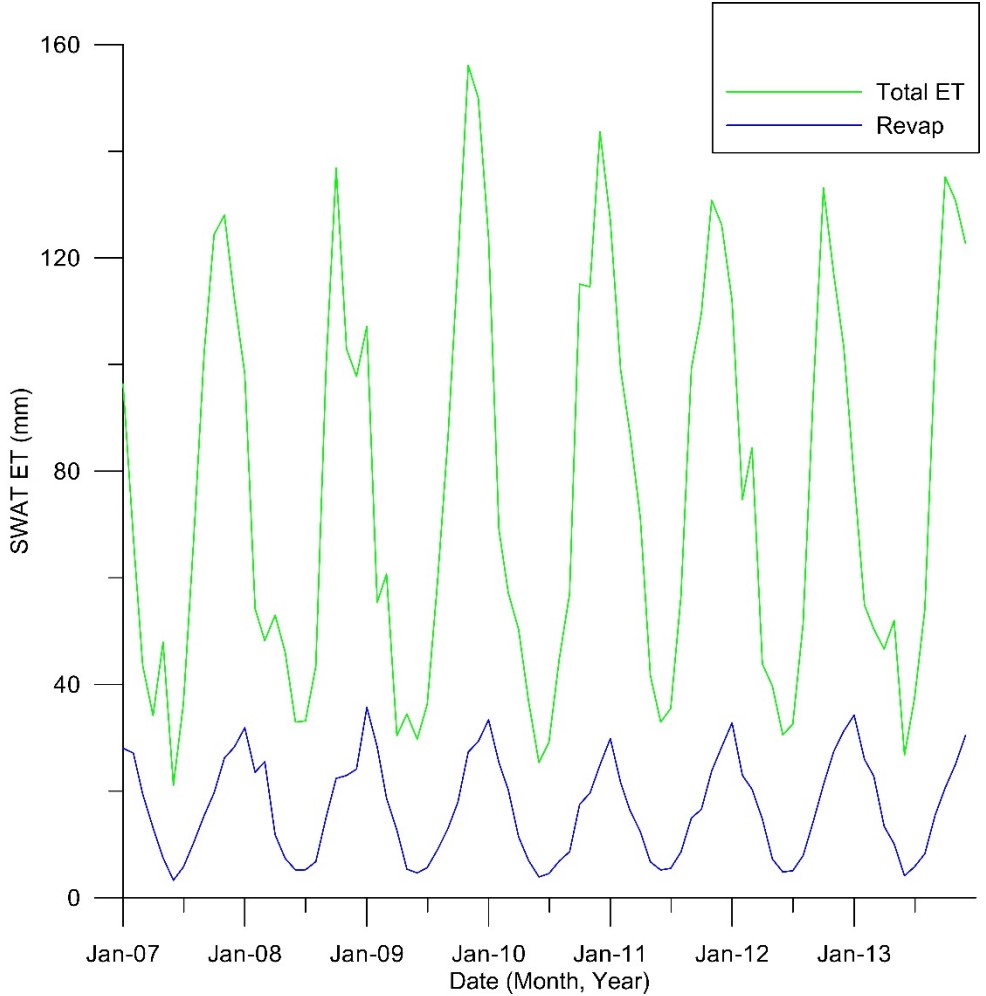


**Fig 10. Monthly comparison of Revap component of the ET and total ET in SWAT.**



On catchment scale, the results show that MOD16 simulates higher ET in the winter periods while SWAT
simulates higher ET during the summer periods (Fig. 9). Generally, the agreement between the products is more
consistent during the winter seasons when ET is lower. The lesser correlation during higher ET seasons may be
related to the linearly determined Revap component of the ET, which is a more dominant process in the summer
months when the demand for soil evaporation, plant transpiration and groundwater ET is significantly higher.
**5.2.3     HRU parameterization in AWRA-L**

The HRU parameterization method in AWRA-L significantly impacts the evapotranspiration modelling process.
While the AWRA-L does not use a robust land cover product that distinguishes between vegetation including
trees, it uses a fraction of tree cover product to parameterise the HRU. AWRA-L discretises each 5 km$^2$ grid cell
into two HRU's; the shallow-rooted HRU and the deep-rooted HRU. The determination of the area of the grid
apportioned as deep-rooted and shallow rooted HRU are solely based on the satellite derived product of the
persistent and recurrent photosynthetically active absorbed radiation ($F_{par}$) from the Advanced Very High
Resolution Radiometer (AVHRR) (Donohue et al., 2008). The fraction of the persistent $F_{par}$ is regarded as the
fraction of tree cover, hence it is used as the fraction of the deep-rooted HRU in each grid cell. The discretisation
of the AWRA-L HRU in the Sixth Creek catchment which suggests under 60% tree cover in the Sixth Creek
Catchment severely limits the access of the model to the deep soil storage and groundwater ET computation in
the catchment, hence the close correlation and agreement of the AWRA-L model with the SWAT model when
the Revap (groundwater ET) is unaccounted for is reasonable.
**5.3   Input data Challenges**
The SWAT ET and the MOD16 methods both have challenges associated with input data, which are subsequently
propagated through the algorithm. In semi-arid environments such as the Sixth Creek Catchment, high intensity
rainfall events are common occurrences, which impacts hydrologic processes such as infiltration and
evapotranspiration differently from if the precipitation were evenly distributed through the day (Syed et al., 2003).
Yang et al. (2016) observed that the use of hourly rainfall in SWAT significantly improved the modelling of
streamflow and hydrological processes. In this study, due to the unavailability of hourly precipitation data, daily
precipitation data were used thus neglecting the impact of high intensity precipitation events in the catchment.

Another challenge encountered with the SWAT model is associated with the semi-distributed model methodology.
The use of a single value for wind speed, relative humidity and solar radiation for a sub-catchment with spatial
scale, which could be in the order of tens of square kilometres, affects the accuracy of hydrological processes at
the HRU scale. The "elevation band" method of temperature and precipitation distribution with respect to
elevation changes across a catchment was introduced into the SWAT algorithm to attenuate orographic effects in
complex terrain catchments (Neitsch et al., 2011). The elevation band algorithm in SWAT has performed well in
predominantly snowy, complex terrain catchments, which are significantly larger than the Sixth Creek Catchment
with elevation changes in the order of kilometres (Abbaspour et al., 2007;Zhang et al., 2008b;Pradhanang et al.,
2011). However, the application of the elevation band algorithm in the non-snowy Odiel River basin (Spain) with
Mediterranean climate similar to the Sixth Creek Catchment yielded less than satisfactory results (Galván et al.,
2014). In the non-snowy Sixth Creek Catchment, the orographic effects are a dominant atmospheric process when
winds are moving from the lower elevations in the north of the catchment to the higher elevations in the South
particularly during the winter months. The orographic lift leads to significantly higher precipitation in the south-
westerly direction in the Sixth Creek Catchment, which the elevation band algorithm in SWAT would not
represent accurately in non-snowy catchments.
The various meteorological and remote sensing input data used in the processing of the MOD16 all have their
inherent uncertainties, with cloud cover challenges and coarse resolution resampling (Mu et al., 2011), while
errors have been associated with the land cover product used (Ruhoff et al., 2013). The land cover map (MOD12)
used in MOD16 (Fig. 5a ), in conjunction with the calibrated biome properties lookup table (BPLUT) significantly
influences the ET output from the various land covers under different climatic conditions. A more detailed map
and local knowledge of the Sixth Creek Catchment indicates that the MOD12 land cover spatially mismatches
some biomes (Fig. 5a and 5b). Besides the obvious land cover mismatches that were observed between the input
data of the two models, the variety of accepted national, regional and global land cover classification system
contributes to the challenges of hydrological modelling. In this MOD12, the "mixed forest" category covered over
50% of the catchment while the category does not exist in the local field map land cover classification. The global
standardization and harmonization of land cover maps and biome classification at high resolution may improve
model performance.

**6   Conclusion**
The main objectives of this paper are to compare three ET products (SWAT, MOD16 and AWRA-L) on catchment
scale, while also evaluating the two finer resolution products (SWAT and MOD16) on graduated spatial scale.
We also attempted to determine the spatial scale at which the models tend towards agreement. while also seeking
to understand the sources of disagreements between the models.

The calibrated SWAT model using the SUFI-2 algorithm and various objective functions could simulate ET to
within 6% of the MOD16 on catchment scale, annually. The P and R factors metrics were observed to be very
reliable indicators of a good calibration exercise. Abbaspour (2007) proposed P and R factor minimum
benchmarks of >0.7 and <1 respectively for streamflow calibration, in this study the P and R factors >0.8 and <1
were found to produce reliable ET estimates on catchment scales. We observed that at a spatial scale of 4 km$^2$ we
obtained cell differences of under 20% annually which gave confidence to our study in the complex terrain that
our 4 km$^2$ aggregation is a good scale of confidence.

The SWAT and MOD16 show good correlation on catchment scale while, the AWRA-L and the SWAT model
without the inclusion of the groundwater ET component of the SWAT model showed good agreement. Biome
differences and input spatial scale contribute to poor agreement at finer spatial scales. The challenge of the lack
of a globally accepted and harmonised land cover classification system at high resolution was encountered in the
study, with two products derived from the MODIS satellite data classifying land cover differently and thus
impacting the results from the SWAT models. The use of different land covers with different classification systems
and parameters are observed to have limited impact on evapotranspiration modelling at coarse spatial resolutions
due to spatial averaging. Nevertheless, the tree cover fraction used in place of a land cover product in the AWRA-
L is also observed to impact the ET modelling, particularly in a groundwater dependent catchment like our study
area. The inherent differences and uncertainties associated with these land cover products will continue to be
propagated through the models, thereby promoting divergence in the drive towards more accurate and finer
resolution evapotranspiration data products. While many concerted research efforts have been made in the past
(Latham, 2009;Friedl et al., 2010), a globally accepted harmonised world land cover database at high resolution
can significantly improve correlation and confidence in high resolution ET products.

The result of the spatial resolution analysis corroborates the view that prevailing ET algorithms and measurement
methods will have certain degree of variability due to the complexity of ET estimation and various drivers of the
contributory processes. The study shows that correlation at catchment scale does not necessarily translate to
correlation at finer spatial scales. The study also highlights the possible challenges of the semi-distributed SWAT
ET algorithm in a complex terrain as the input climate data can be a challenge due to spatial resolution and climate
variability.

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

**Appendix A: Evapotranspiration in SWAT**

SWAT provides the user with three options of modelling ET at the HRU scale and at daily temporal resolution

(Penman-Monteith, Hargreaves or Priestly-Taylor methods). In this study, the Penman-Monteith method is used.

SWAT initially calculates the potential evapotranspiration (PET) for a reference crop (Alfalfa) using the Penman-

Monteith equation for well-watered plants (Jensen et al., 1990):

$$\lambda E_0 = \frac{\Delta(H_{net}-G)+\rho.c_p.\frac{e_{sat}-e}{r_a}}{\Delta+\gamma(1+\frac{r_c}{r_a})} \tag{A1}$$

where $\lambda$ is the latent heat of vaporization (MJ kg$^{-1}$); $E_0$ is the potential evapotranspiration rate (mm/d); $\Delta$ is the

slope of the saturation vapor pressure vs temperature curve (kPa $^{\circ}$C$^{-1}$); $H_{net}$ is the net radiation at the surface (MJ

m$^{-2}$ d$^{-1}$); $G$ is the heat flux density to the ground (MJ m$^{-2}$ d$^{-1}$); $\rho$ is the air density (kg m$^{-3}$); $c_p$ is the specific heat of

dry air at constant pressure (J kg$^{-1}$ K$^{-1}$); P is the atmospheric pressure (kPa); e$_{sat}$ is saturation vapor pressure of air

(kPa); $e$ is water vapor pressure (kPa); $r_a$ is the aerodynamic resistance (s m$^{-1}$); $\gamma$ is the psychometric constant

(kPa $^{\circ}$C$^{-1}$) and $r_c$ is the canopy resistance (s m$^{-1}$).

Total ET (AET) in SWAT is made up of four components: canopy evaporation, transpiration, soil evaporation

and groundwater ET (Revap). Revap is the movement of water from the saturated zone into the overlying

unsaturated zone to supplement the water need for evapotranspiration. The Revap process may be insignificant in

regions where the saturated zone is much deeper than the root zone and as such the result is separately reported

from the ET result in the SWAT result database. As SWAT calculates Revap separately, for a calculation of AET

in regions where the saturated zone is within the root zone, the user should add the Revap result column to the ET

calculations. The AET components are calculated from the PET starting with the canopy evaporation. For this

first component the following storage equations are used in determining the volume of water available for

evaporation from the wet canopy in SWAT

$$C_{day} = C_{mx}\left(\frac{L_{ai}}{L_{ai\_mx}}\right) \tag{A2}$$

when $R'_{day} \leq C_{day} - R_{int(i)}$:

$$R_{int(f)} = R_{int(i)} + R'_{day} ; and\ R_{day} = 0 \tag{A3}$$

when $R'_{day} > C_{day} - R_{int(i)}$:

$$R_{int(f)} = C_{day}; R_{day} = R'_{day} - \left(C_{day} - R_{int(i)}\right) \tag{A4}$$

where $C_{day}$ is the maximum amount of water that can be stored in the canopy on a given day (mm); $C_{mx}$ is the
amount of water that can be stored in the canopy when the canopy is fully matured (mm); $L_{ai}$ is the leaf area index
on a given day (); $L_{ai\_mx}$ is the maximum leaf area index when the plant is fully matured (-); $R_{int(i)}$ is the initial
amount of free water available in the canopy at the beginning of the day (mm); $R_{int(f)}$ is the final amount of free
water available in the canopy at the end of the day (mm); $R'_{day}$ is the amount of precipitation on a given day
before accounting for canopy interception (mm); and $R_{day}$ is the amount of precipitation reaching the soil on a
given day (mm).

The SWAT ET algorithm initially evaporates as much water as can be accommodated in the PET from the wet
canopy. If the total volume of water in canopy storage equals or exceeds PET for the day, then ET is calculated
as;
$$E_a = E_{can} = E_0 \tag{A5}$$
where $E_a$ is AET (mm d$^{-1}$); $E_{can}$ is evaporation from canopy constrained by $E_0$, i.e. PET (mm d$^{-1}$). However, if
the water in canopy storage is less than the PET for the day, transpiration, soil evaporation and Revap are
constrained by $E'_0$, which is the potential evapotranspiration adjusted for the evaporation of the water on the
canopy surface (mm d$^{-1}$).
$$E'_0 = E_0 - E_{can} \tag{A6}$$
The second AET component (transpiration) of SWAT is calculated using the following equations;
$$\lambda E_{t\_max} = \frac{\Delta(H_{net} - G) + \gamma K(\frac{0.622 \cdot \lambda \cdot \rho}{P})\frac{e_{sat} - e}{r_a}}{\Delta + \gamma(1 + \frac{r_c}{r_a})} \tag{A7}$$
$$W_z = \left(\frac{E_{t\_max}}{1 - e^{-\tau}}\right) \times \left(1 - e^{(-\tau \times (\frac{z}{zr}))}\right) \tag{A8}$$
$$W'_l = W_l + (W_d \times e_{pco}) \tag{A9}$$
$$W''_l = W'_l \times e^{\left(5 \times \left(\frac{S_{wl}}{(0.25 \times A_{wcl})} - 1\right)\right)} when\ S_{wl} < 25\%\ of\ A_{wcl} \tag{A10}$$
$$W''_l = W'_l\ when\ S_{wl} > 25\%\ of\ A_{wcl} \tag{A11}$$
$$E_{t,l} = min[W''_l, (S_{wl} - W_{pl})] \tag{A12}$$
$$E_t = \sum_{l=1}^{n} E_{t,l} \tag{A13}$$
where $E_{t\_max}$ is the maximum transpiration rate (mm/d); $K = 8.64 \times 10^4$; P is the atmospheric pressure (kPa);
$W_z$ is the potential water taken up by plant from the soil surface to a specific depth (mm/d) $z$; $\tau$ is the plant water
consumption distribution function; $z$ is the depth from soil surface (mm); $z_r$ is the plant root depth from soil
surface (mm); $W_l$ is the potential water consumption by plant in the soil layer $l$ (mm); $W'_l$ is the potential water
consumption by plant in the layer $l$ adjusted for demand (mm); $W_d$ is the plant water consumption demand deficit
from overlying soil layers (mm); $e_{pco}$ is the plant water consumption compensation factor (-); $W''_l$ is the potential
plant water consumption adjusted for initial soil water content (mm); $S_{wl}$ is the soil water content of layer $l$ in a
day (mm); $A_{wcl}$ is the available water capacity of layer $l$ (mm); $W_{pl}$ is soil water content of layer $l$ at wilting point
(mm); $E_{t,l}$ is the actual transpiration water volume from layer $l$ in a given day (mm/d); $E_t$ is the total actual
transpiration by plants in a given day (mm/d). Plant transpiration parameters such as stomatal conductance,
maximum leaf area index and maximum plant height are retrieved from a SWAT database while climate data
required by the Penman-Monteith method are sourced from input data.

The third AET SWAT component, the soil evaporation on a given day, is a function of the transpiration, degree
of shading and potential evapotranspiration adjusted for canopy evaporation. The maximum soil evaporation on
a given day ($E_s$) (mm d$^{-1}$) is calculated as
$E_s = E'_0 cov_{sol}$                                                                                      (A14)
$cov_{sol} = e^{(-5.0\ 10^{-5}CV)}$                                                                          (A15)
where $cov_{sol}$ is the soil cover index (-) and $CV$ is the aboveground biomass for the day (kg/ha). The maximum
possible soil evaporation in a day is then subsequently adjusted for plant water use ($E'_s$) (mm d$^{-1}$)
$E'_s = min\left(E_s, \frac{E_s\ E'_0}{E_s + E_t}\right)$                                                    (A16)
The SWAT ET algorithm then partitions the evaporative demand between the soils layers, with the top 10 mm of
soil accounting for 50% of soil water evaporated. Equation 17 and 18 are used to calculate the evaporative demand
at specific depths and evaporative demands for soil layers respectively.
$E_{soil,z} = E''_s \frac{z}{z + e^{(2.374 - (0.00713\ z))}}$                                               (A17)
$E_{soil,l} = E_{soil,zl} - E_{soil,zu}.e_{sco}$                                                            (A18)
$E'_{soil,l} = E_{soil,l} \times e^{\left(2.5\times\left(\frac{S_{wl}-F_{cl}}{(F_{cl}-W_{pl})}-1\right)\right)} \ when\ S_{wl} < F_{cl}$     (A19)
$E'_{soil,l} = E_{soil,l}\ when\ S_{wl} > F_{cl}$                                                           (A20)
$E''_{soil,l} = min[E'_{soil,l}, 0.8(S_{wl} - W_{pl})]$                                                     (A21)
$E_{soil} = \sum_{l=1}^{n} E''_{soil,l}$                                                                    (A22)
where $E_{soil,z}$ is the water demand for evaporation at depth $z$ (mm); $E_s''$ is the maximum possible water to be
evaporated in a day (mm); $e_{sco}$ is the soil evaporation compensation factor; $E_{soil,l}$ is the water demand for
evaporation in layer $l$ (mm); $E_{soil,zl}$ is the evaporative demand at the lower boundary of the soil layer (mm);
$E_{soil,zu}$ is the evaporative demand at upper boundary of the soil layer (mm); $F_{cl}$ is the water content of the soil
layer $l$ at field capacity (mm) and $E''_{soil,l}$ is the volume of water evaporated from soil layer $l$ (mm/d); $E_{soil}$ is the
total volume of water evaporated from soil on a given day (mm/d).

The fourth component of the ET calculations in SWAT is referred to as "Revap". Revap in SWAT is the amount
of water transferred from the hydraulically connected shallow aquifer to the unsaturated zone in response to water
demand for evapotranspiration. The Revap component in SWAT is akin to ET from groundwater. Revap is often
a dominant catchment process in a groundwater dependent ecosystem and it is calculated at the HRU scale. Revap
is estimated as a fraction of the potential evapotranspiration (PET) and it is dependent on a threshold depth of
water in the shallow aquifer which is set by the user.
$w_{revap,mx} = \beta_{revap} E_0$ $\qquad$ (A23)
$w_{revap} = w_{revap,mx} - a_{thr}$ $if$
$a_{thr} < a_{sh} < (a_{thr} + w_{revap,mx})$ $\qquad$ (A24)
$w_{revap} = 0$ $\qquad\qquad$ $if \; a_{sh} \leq a_{thr}$ $\qquad$ (A25)
$w_{revap} = w_{revap,mx}$ $\qquad\qquad$ $if \; a_{sh} \geq (a_{thr} + w_{revap,mx})$ $\qquad$ (A26)
where $w_{revap,mx}$ is the maximum volume of water transferred to the unsaturated zone in response to water
shortages for the day (mm); $\beta_{revap}$ is the Revap coefficient (-); $w_{revap}$ is the actual volume of water transferred
to the unsaturated zone to supplement water shortage for the day (mm); $a_{sh}$ is the water volume stored in the
shallow aquifer at the beginning of the day (mm); and the $a_{thr}$ is the threshold water level in the shallow aquifer
required for Revap to occur (mm) (Neitsch et al., 2011).

**Appendix B: MODIS Evapotranspiration**

ET in the MOD16 is a summation of three components: wet canopy evaporation, plant transpiration and soil
evaporation. Wet canopy evaporation ($\lambda_{can}$) in MOD16 is calculated using a modified version of the Penman-
Monteith equation,
$$\lambda E_{can} = \frac{(\Delta H_{net}\ F_C) + \rho c_p (e_{sat} - e)\frac{F_{par}}{r_a} F_{wet}}{\Delta + \left(\frac{P\ C_p\ r_{vc}}{\lambda\ \varepsilon\ r_a}\right)} \qquad (B1)$$
Where the parameters are as earlier defined, $\lambda E_{can}$ is the latent heat flux (Wm$^{-2}$); $H_{net}$ is net radiation relative to
canopy (Wm$^{-2}$); $F_{par}$ is the fraction of absorbed photosynthetically active radiation ; $F_{wet}$ is the fraction of the
soil covered by water; $r_{vc}$ is the resistance to latent heat transfer (s m$^{-1}$); and $\varepsilon$ is the emissivity.

The plant transpiration ($\lambda E_t$) is calculated using another variation of the Penman-Monteith equation,
$$\lambda E_t = \frac{(\Delta H_{net}\ F_C) + \rho c_p (e_{sat} - e)\frac{F_C}{r_a}(1 - F_{wet})}{\Delta + \gamma\left(1 + \frac{r_c}{r_a}\right)} \qquad (B2)$$
The soil evaporation ($\lambda E_{soil}$) is a summation of the potential soil evaporation ($\lambda E_{soil\_POT}$) limited by the soil
moisture constraint function (Fisher et al., 2008) and the evaporation from wet soil ($\lambda E_{wet\_soil}$):
$$\lambda E_{soil} = \lambda E_{wet\_soil} + \lambda E_{soil\_POT}\left(\frac{R_h}{100}\right)^{\frac{V_{PD}}{\phi}} \qquad (B3)$$
$$\lambda E_{wet\_soil} = \frac{(\Delta\ H_{net}) + \rho c_p (1.0 - F_C)\frac{V_{PD}}{r_a}(F_{wet})}{\Delta + \gamma\left(\frac{r_{tot}}{r_a}\right)} \qquad (B4)$$
$$\lambda E_{soil\_POT} = \frac{(\Delta H_{net}) + \rho c_p (1.0 - F_C)\frac{V_{PD}}{r_a}(1 - F_{wet})}{\Delta + \gamma\left(\frac{r_{tot}}{r_a}\right)} \qquad (B5)$$
where $H_{net}$ and $r_a$ are relative to the soil surface; $r_{tot}$ is the total aerodynamic resistance to vapor transport (s m$^{-1}$
$^{-1}$); $V_{PD}$ is the vapor pressure deficit (Pa); $R_h$ is the relative humidity (%); and β is a dimnesionless coefficient
defining the relative sensitivity of $R_h$ to $V_{PD}$. In MOD16 the constant $\phi$ is set to 200.
Total evapotranspiration ($\lambda E$) in MOD16 is thus calculated as
$$\lambda E = \lambda E_{can} + \lambda E_t + \lambda E_{soil} \qquad (B6)$$
