# Peer review of "Comparison of MODIS and SWAT Evapotranspiration over"

_Hydrology and Earth System Sciences, 2017_

## Referee Comment (RC1) · Anonymous Referee #1 · 30 Nov 2017

This study compares evapotranspiration (ET) estimates derived from two reknown methods, namely the MOD16 processor and the SWAT model (using the Penman Monteith module for ET). Comparisons are made over the 44 km2 Sixth Creek Catchment of South Australia at different spatial scales, ranging from 1 km2 to 41 km2 resolution.

The paper is well structured and written. My main concern is about the interpretation of the obtained discrepancies between MOD16 and SWAT ET estimates. MOD16 and SWAT are certainly very different approaches, but they have in common several input variables including the land cover and meteorological forcings. It is therefore regrettable that the authors did not use the MOD16 input data set for their SWAT simulations.

[Figure]

Not because one input data set is more accurate than the other, but rather because it would have been a way to understand and quantitatively assess the origin of the observed differences between MOD16 and SWAT ET estimates. Another drawback is that there is no reference (e.g. in situ meausrements or reference model runs) for evaluating the comparison between MOD16 and SWAT ET estimates. Therefore, it is difficult to assess the significance of either MOD16 or SWAT ET estimates, especially at the 1 km resolution.

Major issues : 1) To me "the drivers of the ET algorithm in both models" (one main objective of the paper, stated at line 420) are not evaluated quantitatively. Abstract, Lines 18-20 : "Land cover differences, mismatches between the two methods and catchment-scale averaging of input data in the SWAT semi-distributed model were identified as the principal sources of weaker correlations at higher spatial resolution". As different data sets were used as input to both MOD16 and SWAT, the above statement is rather an assumption than an "identification". A sensitivity analysis of SWAT model to different forcings (including the MOD16 forcing data) is needed.

2) Figure 7 : I am concerned about the significance of the results at 20 and 41 km2 due to the limited extent of the study area. At those scales (which are about the size of the catchment), the differences in ET estimation are attributed to time only, while at the 1 km2 resolution, the differences in ET estimation are attributed to both space and time. Therefore, those statistics are not, strictly speaking, comparable. It is necessary to separate the spatial differences from the temporal differences at all spatial scales. Otherwise no firm conclusion can be drawn. In addition, since aggregation systematically reduces variability, it would make sense to plot side-by-side the difference in mm (as already shown in Figure 7) and the % of this difference relative to the mean ET, for each spatial resolution ranging from 1 to 10 km2.

Specific points : - Line 110 : define PET, Ecan, Et, Rsoil and Revap in the text to clarify the schematic diagram of Fig. 1 - Line 311-312 : "The land cover is an important parameter in the MOD16 and SWAT MOD16 ET algorithms as it determines the

values allocated to biophysical properties such as leaf conductance, boundary layer resistance and vapour pressure deficit (VPD)" VPD is rather an atmospheric variable than a surface variable controlled by land cover.
* * *

---

## Referee Comment (RC2) · Anonymous Referee #2 · 11 Jan 2018

Review of the paper " Comparison of MODIS and SWAT Evapotranspiration over a Complex Terrain at Different Spatial Scales", by Olanrewaju O Abiodun, Huade Guan, Vincent E.A. Post, and Okke Batelaan.

The paper deals with the comparison of evapo-transpiration as estimated by the MOD16 satellite product and the SWAT hydrological model. The comparison is made on a 44 km2 semi-arid, mountainous basin in South Australia. The model is calibrated against observed discharged data, and the ET computed with the calibrated model is used for the comparison. The comparison is made on a range of spatial scales from 1 to 40 km2. The authors attempt to analyse the causes of the discrepancies between both products.

**General comments**

The papers addresses an important issue about estimating ET at the basin scale.
The paper is well written, with a clear structure. The language is fluent and precise. The methodology is generally well explained (see the detailed comments below). The paper is not totally novel as Et comparison studies are numerous, but it is  probably novel regarding the region and the kind of basin used.

My main criticism is that the outcome of the paper remain, to me, a bit disappointing as the no clear conclusions are drawn on the cause of the biases between the two products (the discussion section mainly gives assumptions ore statements), nor any hierarchy between the possible causes. I made some suggestions below to help enrich the analyses. I agree, however,  that such "timid" conclusions are inherent to this kind of analyses as the authors can not manipulate the satellite product to really test it, and as no reference ET product is available. May be adding other satellite product to the analysis (ex GLEAM) would be helpful ?

Besides, the authors do not give any informations on the usefulness of the study for other contexts or basins. The discussion may be more elaborated on this aspect,

Further, I do not see the fundamental motivation (obj # 2 of the paper)  to compare both products on graduated spatial scales. I feel it is a way to evaluate the products rather than an objective as such.

The state-of-the-art section should be complemented with references on ET inter-comparison studies (models/satellite/numerical prediction systems), which could also enrich the discussion. See for example (I am not in the author's list !)  Trambauer, et al, 2014, doi:10.5194/hess-18-193-2014. The authors should precise if SWAT has been used in such comparison studies. The novelty of the paper must be better emphasized.

As a conclusion, I think the subject of the paper is interesting but some complimentary analyses are needed before publication to reinforce the scope of the paper.

**Detailed comments**

l 33. the ref to Goyal et al seems not appropriate : this work is a bit old now (2004), and concerns a specific area on Inda. Better cite a/some refs dealing with a global perspective. Moreover I am not convinced that ET will be the most impacted everywhere. May be rainfall or even runoff might be severely affected. Please clarify.

l 50: …. *conducted a review* **OF** ? *30 remote....*

l 53-54 Moran and Jackson 1991 : add more recent reference(s)

Fig 1. Please give the meaning of all the variables (PET, Ecan, Et, Esoil, Revap, ET) in the caption to help understanding the figure.

l. 129 onwards (section 3.1). Some brief elements on the hydrology of the basin would be helpful. Please give an estimate of the mean annual discharge as compared to rainfall. Are the stream ephemeral (as suggested by fig 5), which simplifies the problem of setting the initial conditions for the simulations (see below) ? What is the depth to the ground water (see allusion on line 339). Is ground water the main source of river water ? What is the main sink for groundwater : ET or river discharge ? Impact of pumping ? …..

L 147 : the time range 2002 – 2016 is too short I think to be qualified as "historic"....

l 175 : do you mean : "...  all the 1 km 2 cells that **totally or partially** fall within the catchment area, i.e. you did not weighted the mean by the fraction of the cells overlapping the basin ?

l. 178 : How many HRUs did you use, and in what were their size range ? It could help understanding fig 6a.

L 177 onwards (Section 3.3).

It is unclear to me how you used, practically, the 5 objective functions for calibration, i.e. how you made the best compromise between all of them. Did you use the P- and R-factors metrics only to measure the confidence of the calibration once done, or also to select the parameter sets (as suggested at the end of line 216).

The way you calibrate the model is also unclear to me. It seems that you explored  the parameter space (using Latin Hypercube Sampling)  and selected a range of acceptable values for each parameters (see Table 3) based on the corresponding criteria values. Then which threshold values did you used for each criterion (obj. function ?)

By the way, the KGE criteria includes the three previous ones ( r =R from eq. 4 ; omega = PBIAS from eq. 3 and alpha= Rsr from eq. 2), and is generally considered more robust than NSE,  so I do not clearly understand why you used all 5 ?
Moreover  lines 216-218 suggest that you finally used only P-factor, NSE, Rsr a,nd PBIAS to select the range of parameter values.

Please clarify all these points. May be a little more detailed description of the SUFI-2 algorithm will help understanding the procedure ? Please give a reference for this algorithm.

 L 206-207. Referring to the paper by Gupta et al, (2009),  $r$ is the linear **correlation** (not regression) coefficient between the simulated and measured values

l. 213  : what does PPU stand for ?

L 220. Please detail how you managed the initial conditions at the beginning of each simulations period : spin-up period to equilibrate the internal reservoirs and mass budget of the model, or prescription (e.g. soil moisture, river discharges) from observations, or ... ?

l. 226-227 *"In the SUFI-2 algorithm an "r_" and a "v_" prefix before a SWAT model parameter indicate relative change and replacement change of the actual parameter values, respectively."* : I do not understand what you mean here. Please clarify

l. 240 Table 3. As you come up with a range of value for each parameter after calibration, you get in fact an ensemble of hydrological simulations on each pixel or HRU's and on the whole period. I think the exploitation of this ensemble could be very fruitful, as it gives a kind of uncertainty range on your simulations (see for ex. Beven and Binley, 1992, *doi:10.1002/hyp.3360060305*). The comparison of MOD16 with the ensemble mean +- 1 standard dev, (for ex) would be informative.

l. 249, Table 4. Same comment than above (for section 3.3)  which criterion were actually used for calibration, and what is the added value of R2, Pbias, Rsr as compared to NSE and KGE.

L 257, whole section 4.2.
I do not understand why you only evaluated the differences between the two products (figs 7 and 8), at various spatial scales, and not also (and firstly) their absolute value. It is difficult to clearly figure out which one is higher, where and when.
I suggest you display the empirical, statistical cumulated distributions (cdf) of the 1 km2 ET values for each product (for a given time aggregation period : 8-days, month, year, full period). They will inform on the relative magnitude of each series of value, the position of their means (I expect the mean of each distribution to be close to one another, as the basin-scale ET values are close). This analysis may reveal the causes of the biases. For example if both distribution are "close" or similar . it means that both product generate similar values as a whole but not necessarily over the same pixel or in at the same date (due to a bad land cover in MOD 12, and/or mismatch in the forcing fields...). It could explain, again,  why the basin-scale means are close. I can also inform if this good match occurs for wrong reasons or only by chance ?.
May be the basin-scale averages converge mainly due to energy balance constraints, driven by the Penmann-Monteith equation, as at this spatial scale the atmospheric forcing for each model could be essentially comparable ?

An another informative analysis would be to compare MOD16 and SWAT ET on groups of pixels which correspond to comparable (or the most comparable) land cover  on their respective map (cf fig. 4), even if their location do not match. (e.g. compare MOD16/Grassland and Tussock Grasses+ rainfed pasture ? ). It could help check if a more realistic land cover (MOD12) would have produced a more realistic MOD16 ET.
I think this kind of analyses could enrich the discussion on the cause of the biases in section 5, which are, to me, a bit disappointing as nothing is really proven.

L 258-259 and  Fig 6 b. It seems that you only  considered the 1km2 MODIS cells fully enclosed within the basin boundaries. Please confirm.

L 276 - 208  and fig, 7. Please explain briefly how you aggregated to 2, 5, 10, .. km2 (Did you use the "spatial analyst tool in ArcGIS" as mentioned l 292-293 ?). Pixel grouping (2x2, 3x3, 4x4, ... pixels) results in areas of 4, 9, 16, … km2, which are not the aggregated areas you get. How did you manage the blank, 'no data' zones  which inevitably fall into aggregated pixels ?

lines 278, 303, 428 "correlation" :  except in the legend of fig. 9, it seems to me that you did not explicitly estimated the **correlation** between MOD16 and SWAT ET at each resolution. Please choose an other word or compute the correlations.

And by the way, I am not convinced that a lower max cell difference always indicates a better correlation (as suggested lines 277-279, unless I misunderstood) between the two series (here is a counter-example : consider a random series s(t), and two derived series s1=a.s(t) and s2=b.s(t), where a and b are scalars and a>b ; the correlation between s and s1 and s ans s2 is 1, but max(s1-s) > max(s2-s)). Please rephrase the section in a clearer way.

l. 282, fig 7. Does 41 $km^2$ correspond to the catchment scale ? It is said l. 140 that the basin area is 44 $km^2$. Please clarify.

l. 290, section 4.3.
What about comparing the dynamics at the lowest possible time step (that of MOD16 I guess). You rightly mentioned in the intro of the paper that despite its key role in the water cycle, ET was difficult to assess. Hence I think ET evaluation is also crucial at sub-monthly time steps.

l. 294 : $R^2$ and R are redundant, one of them is sufficient

l. 299, fig 9 (and figs 6, 7). Fig 6 visually suggests that MOD16ET > WAST ET at the catchment scale, which is supported by fig 7 (MOD16-SWAT >0), but which is not obvious from fig 9 where MOD16 ET seems < SWAT ET . Together with the differences MOD16-SWAT, it would be informative you give the absolute values of MOD16 and SWAT ET (in mm), e.g. at the catchment scale, on average on the whole period or year by year (see also my previous comment on that topic)

l. 343-345 : *"The convergence of the results of the two methods is also strongly attributed to the simple averaging …...from the MOD16 and SWAT ET to catchment scales."* Can you give more information to support or demonstrate this statement ? Which alternative averaging method(s) could be used ? Have you tested that they would impact the result at the catchment scale ?

L. 350 *"... deep rooted trees that can access the saturated zone... "* Is it the case on your basin ? Please give the information in the "study area" section.

l. 380-386. A realistic representation of rainfall intensities effectively improves streamflow simulations, specially in semi-arid areas, where Horton runoff dominates. However, but its impact on ET (which is mainly an inter-storm process) is probably weak, or weaker, all the more if you have calibrated the daily discharge, thus ensuring a consistent water balance in the basin. Please give more details on the links you see between high rainfall intensities and ET.

l. 426 *"....reliable ET estimates..."* please provide the basis you used to found this judgement. "Reliable" suggests you have a reference to compare with.....

l 430-431 *"...with two products derived from the MODIS satellite data classifying land cover differently..."* I do not understand : what are the 2 products derived from MODIS (MOD12 and ?)

---

## Author Comment (AC1) · 24 Feb 2018

| Comment 1 | My main concern is about the interpretation of the obtained discrepancies between MOD16 and SWAT ET estimates. MOD16 and SWAT are certainly very different approaches, but they have in common several input variables including the land cover and meteorological forcings. It is therefore regrettable that the authors did not use the MOD16 input data set for their SWAT simulations. |
|---|---|
| **Reply 1** | We acknowledge the concern of the reviewer regarding the interpretation of the discrepancies between the MOD16 and SWAT results. We will therefore rerun the SWAT model with MOD12 Land Cover and a comparison of the results of the difference in the two SWAT model runs relative to the MOD16 will be discussed in the revised manuscript. However, the meteorological forcings, which both model have in common (temperature, humidity and radiation), have a very coarse resolution of $1.0^{\circ}$ x $1.25^{\circ}$ in MOD16, which is significantly larger than our study area. This GMAO (Global Modelling and Assimilation Office) coarse resolution data which was used in the MOD16 product was resampled using a non-linear fourth order cosine function interpolation technique. Moreover, our philosophy of comparison in this manuscript is to evaluate the standard MOD16 against a 'in a typical way' well calibrated SWAT model on catchment scale and not to reduce the quality and accuracy of our SWAT model, which currently use available $0.05^{\circ}$ x $0.05^{\circ}$ and $0.01^{\circ}$ x $0.01^{\circ}$ resolution meteorological data for more accurate results. |
| **Comment 2** | Another drawback is that there is no reference (e.g. in situ measurements or reference model runs) for evaluating the comparison between MOD16 and SWAT ET estimates. Therefore, it is difficult to assess the significance of either MOD16 or SWAT ET estimates, especially at the 1 km resolution. |
| **Reply 2** | The reviewer's comments regarding a lack of in-situ data or reference model in our study area (complex terrain) is one of the motivations for engaging in this work. There is limited work on evapotranspiration in complex terrain due to difficulty of equipment installation and data retrieval. Hence, our attempt to find a way to gain more confidence in evapotranspiration estimates in such terrain leads us to comparing the comprehensively validated and widely used energy-balance based MOD16 and a properly calibrated water-balance based SWAT model. Our rationale of the graduated scale analysis is that if the products begin to agree at a certain spatial scale, then confidence is placed on such an analysis rather than on either model. Also, this may give an indication of what range of scale a degree of confidence can be achieved when using models to determine ET over a complex terrain. |
| **Comment 3** | Major issues : 1) To me "the drivers of the ET algorithm in both models" (one main objective of the paper, stated at line 420) are not evaluated quantitatively. Abstract, Lines 18-20 : "Land cover differences, mismatches between the two methods and catchment scale averaging of input data in the SWAT semi-distributed model were identified as the principal sources of weaker correlations at higher spatial resolution". As different data sets were used as input to both MOD16 and SWAT, the above statement is rather an assumption than an "identification". A |

| | |
|---|---|
| | sensitivity analysis of SWAT model to different forcings (including the MOD16 forcing data) is needed. |
| **Reply 3** | We agree that the drivers of the MOD16 ET algorithm have not been quantitatively evaluated and with the constraint of the differences in meteorological forcings, we concede it may be difficult to quantitatively analyse the drivers of the algorithms. We will therefore include the section 5.1.2 (Line 321) in the methodology section.

We will focus our objectives on;
    1.) To simulate and compare the results of the evapotranspiration of SWAT and MOD16 over a complex terrain in a semi-arid environment on catchment scale
    2.) To analyse and determine the spatial scale at which the SWAT and MOD16 ET models tend towards agreement to enhance confidence in ET estimation in a complex terrain

We will also include the analysis of the land cover differences between both models after the model rerun using MOD12 land cover in SWAT. |
| **Comment 4** | Figure 7 : I am concerned about the significance of the results at 20 and 41 km2 due to the limited extent of the study area. At those scales (which are about the size of the catchment), the differences in ET estimation are attributed to time only, while at the 1 km2 resolution, the differences in ET estimation are attributed to both space and time. Therefore, those statistics are not, strictly speaking, comparable. It is necessary to separate the spatial differences from the temporal differences at all spatial scales. Otherwise no firm conclusion can be drawn. In addition, since aggregation systematically reduces variability, it would make sense to plot side-by-side the difference in mm (as already shown in Figure 7) and the % of this difference relative to the mean ET, for each spatial resolution ranging from 1 to 10 km2 |
| **Reply 4** | We appreciate the reviewer's comment and can see how this will enrich the discussion section and the study as a whole.

In the next version of the manuscript, we will introduce a more rigorous evaluation of the spatial and temporal variance of the two ET results based on the method proposed in Sun et al. (2010) (doi:10.1029/2010GL043323). The grand variance analysis of each of the results will be partitioned into their temporal and spatial variance components. With the spatio-temporal analysis, the causes of the bias will be identified for the various temporal and spatial resolutions. We will include the results in section 4 and the discussion on the spatio-temporal analysis in the section 5 of the manuscript. |
| **Comment 5** | Specific points : - Line 110 : define PET, Ecan, Et, Rsoil and Revap in the text to clarify the schematic diagram of Fig. 1 - |
| **Reply 5** | We have added the definitions to the manuscript |
| **Manuscript Changes 5** | "Where PET is the potential evapotranspiration, Ecan is the evaporation from canopy surface, Et is the transpiration, Esoil is the evaporation from the soil and Revap is the amount of water transferred from the underlying |

| | |
|---|---|
| | shallow aquifer to the unsaturated zone in response to water demand for evapotranspiration." |
| **Comment 6** | Line 311-312 : "The land cover is an important parameter in the MOD16 and SWAT MOD16 ET algorithms as it determines the values allocated to biophysical properties such as leaf conductance, boundary layer resistance and vapour pressure deficit (VPD)" VPD is rather an atmospheric variable than a surface variable controlled by land cover |
| **Reply 6** | This has been rephrased |
| **Manuscript Changes 6** | "The land cover is an important parameter in the MOD16 and SWAT ET algorithms as it determines the values allocated to biophysical properties such as leaf conductance and boundary layer resistance, which significantly impact ET calculations." |

---

## Author Comment (AC2) · 24 Feb 2018

| Comment 1 | My main criticism is that the outcome of the paper remain, to me, a bit disappointing as the no clear conclusions are drawn on the cause of the biases between the two products (the discussion section mainly gives assumptions or statements), nor any hierarchy between the possible causes. I made some suggestions below to help enrich the analyses. I agree, however, that such "timid" conclusions are inherent to this kind of analyses as the authors can not manipulate the satellite product to really test it, and as no reference ET product is available. May be adding other satellite product to the analysis (ex GLEAM) would be helpful ? |
|---|---|
| Reply 1 | We appreciate the reviewer's constructive comment. The philosophy of the paper was to propose a way to gain confidence in ET estimation particularly over complex terrain and at what spatial scale do we begin to have confidence for catchment studies. Hence, the approach was to confront two equal scientific approaches (MOD16 and SWAT) and analyse and compare correspondences and differences. The rationale for this is that several challenges are associated with complex terrain ground-based measurement such as site condition to meet measurement criteria, site access for installation and data retrieval and equipment suitability. Hence, the need to find confidence in currently available hydrological and remote sensing methods on sub-catchment scales.

In view of the above, we will include two paragraphs in the introduction focused on; 1) The complexity of obtaining ET measurements in a complex terrain and 2) The need to gain confidence in a typical ET product based on comparison with other methods; and by analysing for correlation and how dependent the confidence is on the spatial scale.

To achieve the above, we will narrow down our objectives to:
1) To simulate and compare the results of the evapotranspiration of SWAT and MOD16 over a complex terrain in a semi-arid environment on catchment scale.
2) To analyse and determine the spatial scale at which the SWAT and MOD16 ET models tend towards agreement to enhance confidence in scale analysis in a complex terrain

We will also enhance the discussion section by rerunning the SWAT model using the same MOD12 land cover used in MOD16 to enable us analyse the effect of different land covers in the ET estimation in SWAT. Both will be evaluated with the MOD16 product.

We particularly chose the MOD16 ET product due to its wide usage, acceptance and particularly 1 km x 1 km resolution. While the GLEAM is a global product, its 25 km x 25 km resolution does not fit into our catchment scale type of analysis. The Australian Government Bureau of Meteorology however has created another ET product on 5 km x 5 km resolution which has only four cells intersecting our catchment, regardless this, it will be included to enrich the discussion when analysing our results at the catchment scale. |

| Comment 2 | Besides, the authors do not give any information on the usefulness of the study for other contexts or basins. The discussion may be more elaborated on this aspect, |
|---|---|
| **Reply 2** | As noted in the Reply 1 above, the usefulness of the study will be expatiated on in the introduction section, where such spatial scale comparison of two products/ methods using the best available data for both can be compared in complex terrain areas with the aim of increasing the confidence on ET estimation based on the convergence of both methods. Hence, the spatial scale at which they begin to converge (if convergence is observed) will be regarded as the spatial scale of confidence for the study less than 30% difference in accordance with observed differences by ground-based methods on the same site Liu et al. (2013) http://dx.doi.org/10.1016/j.jhydrol.2013.02.025, Mu et al. (2011) http://dx.doi.org/10.1016/j.rse.2011.02.019 ). If no convergence is observed, other products or methods may be included in a multi-method analysis to get confidence around modelled results. |
| Comment 3 | Further, I do not see the fundamental motivation (obj # 2 of the paper) to compare both products on graduated spatial scales. I feel it is a way to evaluate the products rather than an objective as such. |
| **Reply 3** | This objective has been refined as below: |
| **Manuscript Changes 3** | 1) To analyse and determine the spatial scale at which the SWAT and MOD16 ET models tend towards agreement to enhance confidence in scale analysis in a complex terrain. |
| Comment 4 | The state-of-the-art section should be complemented with references on ET inter-comparison studies (models/satellite/numerical prediction systems), which could also enrich the discussion. See for example (I am not in the author's list !) Trambauer, et al, 2014, doi:10.5194/hess-18-193-2014. The authors should precise if SWAT has been used in such comparison studies. The novelty of the paper must be better emphasized. |
| **Reply 4** | The above study has been added to Table 1 which details several ET inter-comparison studies. To the best of the authors' knowledge SWAT has been used in just a single ET inter-comparison study (Gao and Long 2008, **DOI:** 10.1002/hyp.7104). |
| Comment 5 | the ref to Goyal et al seems not appropriate : this work is a bit old now (2004), and concerns a specific area on Inda. Better cite a/some refs dealing with a global perspective. Moreover I am not convinced that ET will be the most impacted everywhere. May be rainfall or even runoff might be severely affected. Please clarify. |
| **Reply 5** | This has been rephrased as |
| **Manuscript Changes 5** | "Moreover, ET will be one of the most severely impacted hydrological components of the water cycle alongside precipitation and runoff as a consequence of global climate change (Abtew and Melesse 2013)." DOI https://doi.org/10.1007/978-94-007-4737-1_13 |
| Comment 6 | …. *conducted a review* **OF** ? *30 remote....* |
| **Reply 6** | This has been corrected to read |

| | |
|---|---|
| **Manuscript Changes 6** | "…conducted a review of 30 remote…." |
| **Comment 7** | l 53-54 Moran and Jackson 1991 : add more recent reference(s) |
| **Reply 7** | More recent references have been added to the manuscript |
| **Manuscript Changes 7** | "..(Moran and Jackson, 1991;Verstraeten et al., 2008; Melesse et al., 2009; Fernandes et al., 2012)." |
| **Comment 8** | Fig 1. Please give the meaning of all the variables (PET, Ecan, Et, Esoil, Revap, ET) in the caption to help understanding the figure. |
| **Reply 8** | We have added the definitions below to the manuscript |
| **Manuscript Changes 8** | "Where PET is the potential evapotranspiration, $E_{can}$ is the evaporation from canopy surface, $E_t$ is the transpiration, $E_{soil}$ is the evaporation from the soil and Revap is the amount of water transferred from the underlying shallow aquifer to the unsaturated zone in response to water demand for evapotranspiration." |
| **Comment 9** | l. 129 onwards (section 3.1). Some brief elements on the hydrology of the basin would be helpful. Please give an estimate of the mean annual discharge as compared to rainfall. Are the stream ephemeral (as suggested by fig 5), which simplifies the problem of setting the initial conditions for the simulations (see below) ? What is the depth to the ground water (see allusion on line 339). Is ground water the main source of river water ? What is the main sink for groundwater : ET or river discharge ? Impact of pumping ? ….. |
| **Reply 9** | We have added the following to the manuscript |
| **Manuscript Changes 9** | The Sixth Creek is a perennial stream with mean annual discharge of 0.25 $m^3$/s which accounts for 20–25 % of the mean annual rainfall in the catchment. The Sixth Creek did however experience a total of 35 days of no flow in the 13 year period of this study which encompasses the "millennium drought years" (2000 – 2009) in Australia. The Sixth Creek is a gaining stream with groundwater discharging into the stream and sustaining it especially during the dry summer months. The depth to groundwater varies greatly across the complex terrain catchment, from less than 1 m to over 20 m across the seasons. |
| **Comment 10** | L 147 : the time range 2002 – 2016 is too short I think to be qualified as "historic".... |
| **Reply 10** | This has been rephrased in the manuscript |
| **Manuscript Changes 10** | The Sixth Creek Catchment's complex terrain plays a significant role in its hydrology, with highly localised precipitation events recorded from the two weather stations in the catchment throughout the 13-year study period (2000 – 2005 and 2007 – 2013). |
| **Comment 11** | do you mean : "... all the 1 km 2 cells that **totally or partially** fall within the catchment area, i.e. you did not weighted the mean by the fraction of the cells overlapping the basin ? |
| **Reply 11** | I used only cells that fell totally into the catchment, hence the 41 km2 and not the catchment size of 44 km2. However, we have decided to use a weighted mean to encompass the cells overlapping the basin in the revised version of the manuscript. |

| | |
|---|---|
| **Comment 12** | l. 178 : How many HRUs did you use, and in what were their size range ? It could help understanding fig 6a. |
| **Reply 12** | 124 HRU's, ranging from 0.001 km2 to 6 km2. However, the HRU's such as the 6 km2 is made up of multiple small areas with the same soil, slope and landcover summed up together across the catchment. Not a single block of 6 km2, hence, a few thousand polygons make up the 124 HRU's.

Below change has been added to the manuscript |
| **Manuscript Changes 12** | The soil, land cover and DEM derived slope data were classified into classes and used to create 124 unique HRU's ranging from 0.001 km$^2$ to 6 km$^2$ in area. While each unique HRU has specific set of properties several small areas with the same land cover, slope and soil type make up the total area of a single HRU. |
| **Comment 13** | L 177 onwards (Section 3.3).
It is unclear to me how you used, practically, the 5 objective functions for calibration, i.e. how you made the best compromise between all of them. Did you use the P- and R-factors metrics only to measure the confidence of the calibration once done, or also to select the parameter sets (as suggested at the end of line 216). |
| **Reply 13** | A clearer explanation of this will be adapted into the manuscript.
Only a single objective function is used for the calibration, i.e. NSE, however the SWAT-CUP calculates the result of the other objective functions and outputs it. The P and R factor metric are also calculated at the end of the "best simulation" based on the objective function used. i.e. for instance, a high NSE may be obtained but could still correspond to a poor P and R factor; the user then determines such to be an unrealistic result, as the NSE does not encompass most of the observations. In such an instance, using the sensitivity analysis result of the calibrated parameters, the user will adjust the parameters within the realistic range for each parameter and rerun the model until a combination of a good P and R factor, which essentially encompasses the observation, aligns with a good NSE. Once such is achieved, the parameters are no longer changed then the model will be set to run for the validation period with the same parameters. This validation result is then compared to observed data. It is this validation outcome that is presented in our study for the years 2007 to 2013.

The following has been added to the manuscript: |
| **Manuscript Changes 13** | For streamflow calibration and validation to be considered reliable, combined satisfactory values should be obtained of P-factor ($> 0.7$), R-factor ($< 1$) (Abbaspour, 2007) and of one of the objective functions, $N_{SE}$ ($> 0.5$), $R_{SR}$ ($\leq 0.7$) and $P_{BIAS}$ ($\pm 25\%$) (Moriasi et al., 2007). In this study, the NSE objective function combined with the P and R factors are used. The result of the other objective functions at the optimal NSE are also recorded. For a comprehensive explanation of the SUFI-2 algorithm, see Abbaspour (2007). |
| **Comment 14** | The way you calibrate the model is also unclear to me. It seems that you explored the parameter space (using Latin Hypercube Sampling) and |

| | |
|---|---|
| | selected a range of acceptable values for each parameters (see Table 3) based on the corresponding criteria values. Then which threshold values did you used for each criterion (obj. function ?). |
| **Reply 14** | A realistic range of values for the set of parameters are calibrated using the Latin Hypercube Sampling method, using only the NSE objective function with a threshold value of 0.5. The model for instance will explore the spaces in the parameters in 500 runs and the best NSE value obtained in the model runs is presented as the "optimal" run. The user then determines if this calibration is good enough (based on the P- and R-values) before proceeding to validation. The guidelines for the calibration set out in Moriasi et al. (2007) and Abbaspour (2007) were the benchmark in this study. |
| **Comment 15** | By the way, the KGE criteria includes the three previous ones ( r = R from eq. 4 ; omega = PBIAS from eq. 3 and alpha= Rsr from eq. 2), and is generally considered more robust than NSE, so I do not clearly understand why you used all 5 ? Moreover lines 216-218 suggest that you finally used only P-factor, NSE, Rsr and PBIAS to select the range of parameter values. |
| **Reply 15** | The text has been edited to reflect clearly the used objective function in Reply 20 and Manuscript Change 19

 The NSE was the principal objective function used in the model, while the P and R factors were monitored to ensure the model encompassed most of the observations. The final simulations used however, corresponded to KGE of 0.71 in calibration and 0.88 in validation on daily time scale |
| **Manuscript Changes 15** | This semi-automatic Latin Hypercube Sampling algorithm optimizes SWAT model parameters while attempting to fit the simulated data as close as possible to the observed data using the user preferred objective function from those detailed below as measurement of simulation accuracy (Abbaspour, 2007). Although a single user objective function is used in the calibration and validation, the results of the other objective functions are also recorded for the optimal model run. |
| **Comment 16** | Please clarify all these points. May be a little more detailed description of the SUFI-2 algorithm will help understanding the procedure ? Please give a reference for this algorithm. |
| **Reply 16** | A reference has been given for the SUFI-2 algorithm in the manuscript (Abbaspour 2007) |
| **Manuscript Changes 16** | In this study, the NSE objective function combined with the P and R factors are used. The result of the other objective functions at the optimal NSE are also recorded. For a comprehensive explanation of the SUFI-2 algorithm, see Abbaspour (2007). |
| **Comment 17** | L 206-207. Referring to the paper by Gupta et al, (2009), $r$ is the linear **correlation** (not regression) coefficient between the simulated and measured values |
| **Reply 17** | This has been corrected |
| **Manuscript Changes 17** | "where $r$ is the linear correlation coefficient between the simulated and measured variable" |
| **Comment 18** | l. 213 : what does PPU stand for ? |

| | |
|---|---|
| **Reply 18** | Line 213 and 214 (Percent prediction uncertainty). This has been added to the manuscript |
| **Manuscript Changes 18** | The P-factor which is also referred to as the 95 Percent Prediction Uncertainty (95PPU), is the percentage of observed data captured which falls between the 2.5 and 97.5 percentiles, while the R-factor is the width of the 95PPU. |
| **Comment 19** | L 220. Please detail how you managed the initial conditions at the beginning of each simulations period : spin-up period to equilibrate the internal reservoirs and mass budget of the model, or prescription (e.g. soil moisture, river discharges) from observations, or ... ? |
| **Reply 19** | The SWAT model encourages the use of a "warm up" period for equilibration, in our model we used a 5 year "warm up" period from 1995 to 1999, which is not included in the calibration or validation periods. |
| **Manuscript Changes 19** | A warm up period of 5 years between 1995 and 1999 was used in the SWAT model to equilibrate the model mass budget and internal reservoirs. |
| **Comment 20** | l. 226-227 "*In the SUFI-2 algorithm an "r_" and a "v_" prefix before a SWAT model parameter indicate relative change and replacement change of the actual parameter values, respectively.*" : I do not understand what you mean here. Please clarify |
| **Reply 20** | This has been clarified in the manuscript

V__ means an existing parameter value to be replaced by a user specified value

While r__ means an existing parameter value to be multiplied by (1 + a user specified value)

The groundwater related parameter were the ones principally replaced. |
| **Manuscript Changes 20** | In the SUFI-2 algorithm, an "r_" and a "v_" prefix before a SWAT model parameter (Table 3) indicate a relative change (a percentage increase or decrease in the SWAT modelled value) and replacement change of the original SWAT modelled values respectively. The relative change is used for fine tuning a model parameter already within an acceptable range, while the replacement change is used when a model parameter is different from measured field parameter or other knowledge base. |
| **Comment 21** | l. 240 Table 3. As you come up with a range of value for each parameter after calibration, you get in fact an ensemble of hydrological simulations on each pixel or HRU's and on the whole period. I think the exploitation of this ensemble could be very fruitful, as it gives a kind of uncertainty range on your simulations (see for ex. Beven and Binley, 1992, *doi:10.1002/hyp.3360060305*). The comparison of MOD16 with the ensemble mean +- 1 standard dev, (for ex) would be informative. |
| **Reply 21** | This is an idea which is worth exploring in the future. However, the structure of the SUFI-2 algorithm somewhat makes this process difficult and cumbersome and almost impracticable as it would require manually attempting to copy information of 124 HRU's in 1000 model simulations. As SUFI-2 is a proprietary software with licensing, automating this process through modifying the software is beyond the scope of this work. |

| | |
|---|---|
| **Comment 22** | l. 249, Table 4. Same comment than above (for section 3.3) which criterion were actually used for calibration, and what is the added value of R2, Pbias, Rsr as compared to NSE and KGE. |
| **Reply 22** | This has been answered in Reply 13, 14, 15 and 16 |
| **Comment 23** | L 257, whole section 4.2. I do not understand why you only evaluated the differences between the two products (figs 7 and 8), at various spatial scales, and not also (and firstly) their absolute value. It is difficult to clearly figure out which one is higher, where and when. I suggest you display the empirical, statistical cumulated distributions (cdf) of the 1 km2 ET values for each product (for a given time aggregation period : 8-days, month, year, full period). They will inform on the relative magnitude of each series of value, the position of their means (I expect the mean of each distribution to be close to one another, as the basin-scale ET values are close). This analysis may reveal the causes of the biases. For example if both distribution are "close" or similar, it means that both product generate similar values as a whole but not necessarily over the same pixel or in at the same date (due to a bad land cover in MOD 12, and/or mismatch in the forcing fields...). It could explain, again, why the basin-scale means are close. I can also inform if this good match occurs for wrong reasons or only by chance ?. May be the basin-scale averages converge mainly due to energy balance constraints, driven by the Penmann-Monteith equation, as at this spatial scale the atmospheric forcing for each model could be essentially comparable ? |
| **Reply 23** | We appreciate the suggestions and see how this will enrich the discussion section. In the next version of the manuscript, we will introduce a more rigorous evaluation of the spatial and temporal variance of the two ET results based on the method proposed in Sun et al. (2010) (doi:10.1029/2010GL043323). The grand variance analysis of each of the results will be partitioned into their temporal and spatial variance components. With the spatio-temporal analysis, the causes of the bias will be identified for the various temporal and spatial resolutions. We will include the results in section 4 and the discussion on the spatio-temporal analysis in the section 5 of the manuscript. |
| **Comment 24** | An another informative analysis would be to compare MOD16 and SWAT ET on groups of pixels which correspond to comparable (or the most comparable) land cover on their respective map (cf fig. 4), even if their location do not match. (e.g. compare MOD16/Grassland and Tussock Grasses+rainfed pasture ? ). It could help check if a more realistic land cover (MOD12) would have produced a more realistic MOD16 ET. I think this kind of analyses could enrich the discussion on the cause of the biases in section 5, which are, to me, a bit disappointing as nothing is really proven. |
| **Reply 24** | We will rerun the SWAT model with MOD12 and we will discuss the results of the SWAT model with the MOD12 land cover and the Geoscience Australia land cover and cross analyse with the MOD16 in section 5 to give a more robust analysis. The section will also include the ET over sections with similar land cover (Geoscience vs MOD12). This will help shed further light on the effect of the land cover on the ET estimations. |

| | |
|---|---|
| **Comment 25** | L 258-259 and Fig 6 b. It seems that you only considered the 1km2 MODIS cells fully enclosed within the basin boundaries. Please confirm. |
| **Reply 25** | Yes I only presented/considered 1 km2 cells fully enclosed in the basin boundaries after discarding the data of those not fully enclosed |
| **Comment 26** | L 276 - 208 and fig, 7. Please explain briefly how you aggregated to 2, 5, 10, .. km2 (Did you use the "spatial analyst tool in ArcGIS" as mentioned l 292-293 ?). Pixel grouping (2x2, 3x3, 4x4, ... pixels) results in areas of 4, 9, 16, … km2, which are not the aggregated areas you get. How did you manage the blank, 'no data' zones which inevitably fall into aggregated pixels ? |
| **Reply 26** | This analysis will be reworked in both SWAT models. The Zonal statistics tool in the spatial analyst toolbox is used in the analysis. The reviewer's pixel grouping suggestion is a better method than previously used and will be applied giving the analysis in 1, 4, 9, 16 and 25 km2 for both SWAT models. Every pixel not fully enclosed within the boundaries of the catchment is weighted less than 100% (according to the percentage of the pixel falling within the catchment) in the zonal statistics analysis. Hence, a greater weight is placed on cells totally enclosed in the catchment. |
| **Comment 27** | lines 278, 303, 428 "correlation" : except in the legend of fig. 9, it seems to me that you did not explicitly estimated the **correlation** between MOD16 and SWAT ET at each resolution. Please choose an other word or compute the correlations. |
| **Reply 27** | These correlations will be computed after the new model run. |
| **Comment 28** | And by the way, I am not convinced that a lower max cell difference always indicates a better correlation (as suggested lines 277-279, unless I misunderstood) between the two series (here is a counter-example : consider a random series s(t), and two derived series s1=a.s(t) and s2=b.s(t), where a and b are scalars and a>b ; the correlation between s and s1 and s ans s2 is 1, but max(s1-s) > max(s2-s)). Please rephrase the section in a clearer way. |
| **Reply 28** | While this has been rephrased in the manuscript as shown below, further analysis will be included after the model runs including the correlations at different spatial resolution. |
| **Manuscript Changes 28** | "Figure 7 show that with increased cell aggregation, the percentage maximum difference between the two methods narrowed. At 5 km2 resolution, the maximum mean difference between the methods narrowed to 21% from a maximum of 48% at 1 km2." |
| **Comment 29** | l. 282, fig 7. Does 41 km2 correspond to the catchment scale ? It is said l. 140 that the basin area is 44 km2. Please clarify. |
| **Reply 29** | The review on the graph will have the 41 km2 replaced with Catchment scale and will be updated as same. |
| **Comment 30** | l. 290, section 4.3. What about comparing the dynamics at the lowest possible time step (that of MOD16 I guess). You rightly mentioned in the intro of the paper that despite its key role in the water cycle, ET was difficult to assess. Hence, I think ET evaluation is also crucial at sub-monthly time steps. |
| **Reply 30** | Although MOD16 is available at a 8-day time resolution we have in this research deliberately chosen to work on a monthly time resolution for the comparison with SWAT. The reason is that evaluation of many |

| | (complex) catchments, which often have limited measured data, is still mostly at monthly time scales. Moreover, temporal patterns of ET over a number of years is also appropriately evaluated at monthly time resolution. Hence, we believe that in accordance with the goals of this paper an evaluation at monthly resolution is fitting. |
|---|---|
| | However, we acknowledge that there is of course interesting sub-monthly variation possibly present. We will add a discussion on this as well as provide recommendations for future research to investigate these sub-monthly ET dynamics. |
| **Comment 31** | l. 294 : R2 and R are redundant, one of them is sufficient |
| **Reply 31** | This has been edited in the manuscript |
| **Manuscript Changes 31** | "Using the $R_{MSE}$, $M_D$ and R$^2$ metrics the analysis shows…" |
| **Comment 32** | l. 299, fig 9 (and figs 6, 7). Fig 6 visually suggests that MOD16ET > WAST ET at the catchment scale, which is supported by fig 7 (MOD16-SWAT >0), but which is not obvious from fig 9 where MOD16 ET seems < SWAT ET . Together with the differences MOD16-SWAT, it would be informative you give the absolute values of MOD16 and SWAT ET (in mm), e.g. at the catchment scale, on average on the whole period or year by year (see also my previous comment on that topic) |
| **Reply 32** | Towards catchment scales the values are closer together actually, Fig 9 still has MOD16 higher (the mean annual between 2007 – 2013 for MOD16 is 916 mm while SWAT ET is 908 mm ). The yearly absolutes will be included alongside the results of the new SWAT model. |
| **Comment 33** | l. 343-345 : *The convergence of the results of the two methods is also strongly attributed to the simple averaging …..from the MOD16 and SWAT ET to catchment scales."* Can you give more information to support or demonstrate this statement ? Which alternative averaging method(s) could be used ? Have you tested that they would impact the result at the catchment scale ? |
| **Reply 33** | Other possible methods which were not included in the manuscript bilinear interpolation, cubic interpolation and nearest neighbour interpolation when going from 1 km2 to catchment scale aggregation. A reference and discussion on these methods and our choice of method will be included in the revised manuscript |
| **Comment 34** | L. 350 *"... deep rooted trees that can access the saturated zone... "* Is it the case on your basin ? Please give the information in the "study area" section. |
| **Reply 34** | Yes this conclusion is based on the history of the native eucalyptus trees in the catchment. I have added some information on the study area in the manuscript. |
| **Manuscript Changes 34** | "The land cover consists of 95% native forestland with significant deep rooted Eucalyptus plantation and 5% pasture, shrubs and grasslands (Fig. 4b). Most of the native vegetation is under conservation". |
| **Comment 35** | l. 380-386. A realistic representation of rainfall intensities effectively improves streamflow simulations, specially in semi-arid areas, where Horton runoff dominates. However, but its impact on ET (which is mainly an inter-storm process) is probably weak, or weaker, all the more if you have calibrated the daily discharge, thus ensuring a consistent |

| | water balance in the basin. Please give more details on the links you see between high rainfall intensities and ET. |
|---|---|
| **Reply 35** | The effect of high intensity rainfall on ET will best be analysed on sub daily and daily timescales. While we can analyse this for the SWAT ET, it is not possible to analyse effectively for the MOD16 due to its coarse temporal resolution. We feel this may lead to the analysis appearing as an isolated discussion in the manuscript as it does not cover both methods under analysis. |
| **Comment 36** | l. 426 *"....reliable ET estimates..."* please provide the basis you used to found this judgement.
"Reliable" suggests you have a reference to compare with..... |
| **Reply 36** | This has been rephrased |
| **Manuscript Changes 36** | "…comparable ET estimates with the MOD16 on catchment scale" |
| **Comment 37** | l 430-431 *"...with two products derived from the MODIS satellite data classifying land cover differently..."* I do not understand : what are the 2 products derived from MODIS (MOD12 and ?) |
| **Reply 37** | The MOD12 and the Geoscience Australia Dynamic Land Cover product are both derived from the MODIS satellite data. However, the Geoscience Australia 250 m Land Cover used in the SWAT model is derived from the MOD13 Enhanced Vegetation Index product. |